# Convergence of Regret Matching in Potential Games and Constrained Optimization

**Ioannis Anagnostides**
Carnegie Mellon University
ianagnos@cs.cmu.edu

**Emanuel Tewolde**
Carnegie Mellon University
etewolde@cs.cmu.edu

**Brian Hu Zhang**
MIT
zhangbh@csail.mit.edu

**Ioannis Panageas**
University of California, Irvine
ipanagea@ics.uci.edu

**Vincent Conitzer**
Carnegie Mellon University
University of Oxford
conitzer@cs.cmu.edu

**Tuomas Sandholm**[*]
Carnegie Mellon University
sandholm@cs.cmu.edu

## ABSTRACT

*Regret matching (RM)*—and its modern variants—is a foundational online algorithm that has been at the heart of many AI breakthrough results in solving benchmark zero-sum games, such as poker. Yet, surprisingly little is known so far in theory about its convergence beyond two-player zero-sum games. For example, whether regret matching converges to Nash equilibria in *potential games* has been an open problem for two decades. Even beyond games, one could try to use RM variants for general constrained optimization problems. Recent empirical evidence suggests that they—particularly *regret matching$^+$ (RM$^+$)*—attain strong performance on benchmark constrained optimization problems, outperforming traditional gradient descent-type algorithms.

We show that RM$^+$ converges to an $\epsilon$-KKT point after $O_\epsilon(1/\epsilon^4)$ iterations, establishing for the first time that it is a sound and fast first-order optimizer. Our argument relates the KKT gap to the accumulated *regret*, two quantities that are entirely disparate in general but interact in an intriguing way in our setting, so much so that when regrets are bounded, our complexity bound improves all the way to $O_\epsilon(1/\epsilon^2)$. From a technical standpoint, while RM$^+$ does *not* have the usual one-step improvement property in general, we show that it does in a certain region that the algorithm will quickly reach and remain in thereafter. In contrast, our second main result establishes that RM, with or without alternation, can take an exponential number of iterations to reach a crude approximate solution even in two-player potential games. This represents the first worst-case separation between RM and RM$^+$. Our lower bound shows that convergence to coarse correlated equilibria in potential games is exponentially faster than convergence to Nash equilibria.

## 1 INTRODUCTION

*Regret matching* is a foundational online algorithm for minimizing *regret*. It was famously introduced by Hart & Mas-Colell (2000), although its conception can be traced much further back to the seminal *approachability* framework of Blackwell (1956), which lay the groundwork for online learning and regret minimization. As the name suggests, regret matching prescribes playing each action with probability proportional to the (nonnegative) regret accumulated by that action. Its appeal lies in its simplicity and scalability, being both *parameter free* and *scale invariant*.

Regret matching—and modern versions thereof—has been at the forefront of equilibrium computation in massive two-player zero-sum games. A notable variant with strong empirical performance is *regret matching$^+$*, introduced by Tammelin (2014); the only difference is that it truncates all negative coordinates of the regret vector to zero in every iteration. Even so, this variant is typically far superior than its predecessor, and was a central component in AI poker breakthroughs (Bowling

---

[*]Additional affiliations: Strategy Robot, Inc., Strategic Machine, Inc., Optimized Markets, Inc.

et al., 2015; Brown & Sandholm, 2017; 2019b; Moravčík et al., 2017) and a more recent superhuman agent for dark chess (Zhang & Sandholm, 2025).

As such, the regret matching family of algorithms has rightfully been the subject of intense study in contemporary research. Much of this focus has been confined to two-player zero-sum games, where minimizing regret translates to convergence—of the *average* strategies—to minimax (equivalently, Nash) equilibria (Freund & Schapire, 1999). More broadly, in general-sum games, no-regret algorithms guarantee convergence to the set of *coarse correlated equilibria* (Moulin & Vial, 1978)—a more permissive concept than Nash equilibria.

In this paper, we examine the convergence of regret matching and its variants in the seminal class of *potential games*, and, more broadly, nonconvex optimization constrained over a product of simplices. Surprisingly little is known about this question even though it was identified early on as an important open question in this space (Kleinberg et al., 2009; Marden et al., 2007). Recent empirical evidence brings this question to the fore again: Tewolde et al. (2025) showed that the regret matching family—and especially regret matching$^+$—attains strong performance on a benchmark suite of constrained optimization problems, significantly outperforming gradient descent-type algorithms. Yet, there is no theory to suggest that regret matching will even asymptotically converge to approximate KKT points in constrained optimization, which are tantamount to Nash equilibria when dealing specifically with potential games. We fill this gap in this paper.

## 1.1 OUR RESULTS

We analyze the convergence of regret matching (RM) and regret matching (RM$^+$) in the general class of (nonconvex) optimization problems constrained over a product of probability simplices. This encompasses as a special case Nash equilibria in potential games when the objective is multilinear. More broadly, to have a unifying treatment of both settings, we think of each probability simplex as being controlled by a single player who is observing the corresponding part of the gradient.

We cover both the simultaneous and the alternating version of RM$^+$—whereby players update their strategies one after the other, akin to coordinate descent. Our result for RM$^+$ is summarized below.

**Theorem 1.1.** *RM$^+$ converges to an $\epsilon$-KKT point of any optimization problem over a product of simplices after $O_\epsilon(1/\epsilon^4)$ iterations.*

This theorem confirms that RM$^+$ is a sound and efficient first-order optimizer, lending further credence to the empirical results of Tewolde et al. (2025). We hope that Theorem 1.1 will help cement RM$^+$ in the optimization arsenal going forward.

We remark that for potential games and constrained optimization over a single simplex, the $O_\epsilon(1/\epsilon^4)$ bound holds no matter how the regrets in (alternating) RM$^+$ are initialized. For constrained optimization over multiple simplices—with or without alternation—we obtain the same rate by suitably initializing the regret vectors (Corollaries 3.11 and C.11); for the usual parameter-free version of RM$^+$, in which the regret vectors are initialized at zero, we can only guarantee an inferior bound growing as $O_\epsilon(1/\epsilon^8)$ (Theorem 3.12).

Our argument proceeds by parameterizing the rate of convergence of RM$^+$ as a function of the accumulated regret, so much so that if the regret with respect to each individual simplex remains bounded, the rate is improved all the way to $T^{-1/2}$.

**Theorem 1.2.** *Suppose that the regret of RM$^+$ on each individual simplex grows as at most $T^\alpha$ for some $\alpha \in [0, 1/2]$. Then RM$^+$ converges to an $\epsilon$-KKT point after $O_\epsilon(1/\epsilon^{2/1-\alpha})$ iterations.*

RM$^+$ always guarantees regret growing as $\sqrt{T}$, so Theorem 1.1 is implied by Theorem 1.2. What makes the latter theorem surprising is that, in general, regret is a fundamentally disparate property compared to KKT gap: as we point out in Proposition 3.2, a sequence can incur zero regret while having an $\Omega(1)$ KKT gap in each iteration. Even so, Theorem 1.2 directly relates the KKT gap in terms of the regret. In particular, the non-asymptotic rate of Theorem 1.1 is a consequence of the fact that RM$^+$ has the no-regret property! In the special case of potential games, regret is known to drive the rate of convergence to *coarse correlated equilibria (CCE)*; Theorem 1.2 shows, for the first time, that regret can also govern the rate of convergence to Nash equilibria. In particular, if convergence to CCE happens at a rate of $T^{-(1-\alpha)}$, for some $\alpha \in [0, 1/2]$, the rate of convergence to Nash equilibria is no slower than $T^{-\frac{1-\alpha}{2}}$.

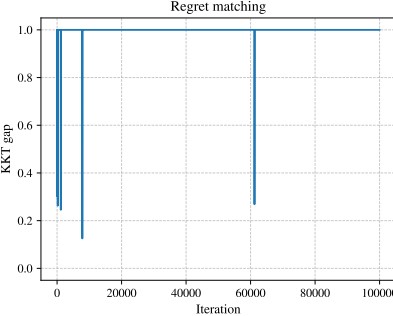 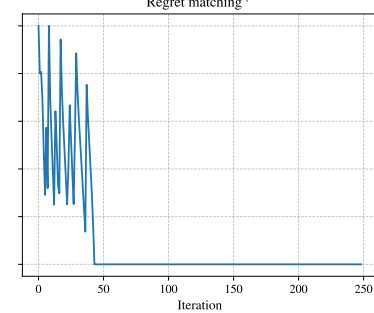

Figure 1: Illustration of our main results: RM$^+$ always converges fast to a KKT point while RM can take exponential time even in two-player identical-interest games, constructed in Section 4.

From a technical standpoint, the key challenge is that RM$^+$ does *not* have a one-step improvement property: even if one initializes RM$^+$ close to a KKT point, RM$^+$ can still grossly overshoot. And, of course, it is a parameter-free algorithm, so the usual treatment of gradient descent-type algorithms that relies on appropriately tuning the learning rate falls short. In this context, our starting observation is that, at least when the utility function is linear, RM$^+$ is bound to improve the utility, although the improvement is inversely proportional to the norm of the regret vector (Lemma 3.3). This key property already suffices to show that alternating RM$^+$ will converge to Nash equilibria in potential games. For the more challenging setting where the updates are simultaneous or the objective is not multilinear, we first show that one-step improvement holds conditional on the norm of the regret vector being *sufficiently large* (Lemmas 3.7 and C.10). To conclude the argument, we combine this property with the crucial insight that the $\ell_2$ norm of the regret vector is *monotonically increasing* proportionally to the KKT gap (Lemma 3.8). This means that RM$^+$ will never get stuck in a cycle: the regret vector would quickly grow in norm, at which point the one-step improvement promised by Lemma 3.7 kicks in.

Does RM share the same convergence properties as RM$^+$? As a reminder, the only difference is that RM refrains from truncating negative regrets to zero. Even so, we find that this seemingly innocuous difference gives rise to an exponential gap in the performance of RM *vis-à-vis* RM$^+$, manifested even in two-player identical-interest games—a special case of potential games (Figure 1).

**Theorem 1.3.** *There is a two-player $m \times m$ identical-interest game where RM, with or without alternation, requires $m^{\Omega(m)}$ iterations to converge to an $m^{-\Theta(1)}$-approximate Nash equilibrium.*

This lower bound holds not just under an adversarial initialization, but also when players initially mix uniformly at random, which is the most common initialization in practice.

Theorem 1.3 constitutes the first worst-case separation—let alone an exponential one—between RM and RM$^+$. Indeed, in zero-sum games, it is known that RM and RM$^+$ both attain a rate no faster than $T^{-1/2}$ (Farina et al., 2023), even though RM$^+$ typically performs much better in practice. Theorem 1.3 provides further justification for opting for RM$^+$ instead of RM, albeit in a fundamentally different setting.

The basic flaw of RM that underpins Theorem 1.3 is that, even with a linear utility, it is not guaranteed to improve the utility even when it has a large best-response gap; specifically, as we show in Lemma 3.6, the improvement is conditional on a good-enough action having nonnegative regret. But herein lies the problem: it could take many iterations before the regret resurfaces to being positive. What happens in the construction behind Theorem 1.3 is that it takes longer and longer—exponentially so—for the regret of the unique good-enough action to be positive; before then, RM is entirely stalled without making any progress. At the same time, RM is guaranteed to converge to the set of coarse correlated equilibria (CCE) at a rate of $T^{-1/2}$, simply because it always has the no-regret property. This leads to the following interesting consequence.

**Corollary 1.4.** *There is a class of two-player potential games in which RM converges to an $\epsilon$-CCE in $O_\epsilon(1/\epsilon^2)$ rounds but it takes $\exp(\Omega(1/\epsilon))$ rounds to converge to an $\epsilon$-Nash equilibrium.*

To be clear, convergence to a CCE is meant in terms of the average correlated distribution of play, whereas convergence to Nash equilibria is in terms of the individual iterates produced by RM.

We defer further discussion on related work to Section A.

## 2 PRELIMINARIES

We begin by introducing potential games and the more general problem of constrained optimization over a product of simplices (Section 2.1), and then recall regret matching($^+$) in Section 2.2.

### 2.1 POTENTIAL GAMES AND CONSTRAINED OPTIMIZATION

**Normal-form games** Our first key focus in this paper is on *potential games*, which we represent in the usual normal form. Here, we have $n$ players, each of whom is to select an action $a_i$ from a finite set $\mathcal{A}_i$, with $m_i := |\mathcal{A}_i|$ and $m = \max_{1 \leq i \leq n} m_i$. Under a joint action profile $\boldsymbol{a} = (a_1, \ldots, a_n) \in \mathcal{A}_1 \times \cdots \times \mathcal{A}_n$, each player $i \in [n]$ receives a payoff given by a *utility function* $u_i : (a_1, \ldots, a_n) \mapsto u_i(a_1, \ldots, a_n) \in \mathbb{R}$ with range bounded by 1. A player $i \in [n]$ can randomize by specifying a *mixed strategy* $\boldsymbol{x}_i \in \Delta(\mathcal{A}_i) := \{\boldsymbol{x}_i \in \mathbb{R}_{\geq 0}^{\mathcal{A}_i} : \sum_{a_i \in \mathcal{A}_i} \boldsymbol{x}_i[a_i] = 1\}$. Player $i$ strives to maximize its *expected* utility, given by $u_i(\boldsymbol{x}_1, \ldots, \boldsymbol{x}_n) := \sum_{(a_1, \ldots, a_n) \in \mathcal{A}_1 \times \cdots \times \mathcal{A}_n} u_i(a_1, \ldots, a_n) \prod_{i'=1}^{n} \boldsymbol{x}_{i'}[a_{i'}]$. A key fact is that the expected utility is *multilinear*, in that $u_i(\boldsymbol{x}_1, \ldots, \boldsymbol{x}_n) = \langle \boldsymbol{x}_i, \boldsymbol{u}_i(\boldsymbol{x}_{-i}) \rangle$ for some utility vector $\boldsymbol{u}_i(\boldsymbol{x}_{-i}) \in \mathbb{R}^{\mathcal{A}_i}$ that does not depend on $\boldsymbol{x}_i$; namely, $\boldsymbol{u}_i(\boldsymbol{x}_{-i}) = (\sum_{(a_1, \ldots, a_{i-1}, a_{i+1}, \ldots, a_n)} u_i(a_1, \ldots, a_n) \prod_{i' \neq i} \boldsymbol{x}_{i'}[a_{i'}])_{a_i \in \mathcal{A}_i}$. Here and throughout, we use the shorthand notation $\boldsymbol{x}_{-i} = (\boldsymbol{x}_1, \ldots, \boldsymbol{x}_{i-1}, \boldsymbol{x}_{i+1}, \ldots, \boldsymbol{x}_n)$, while we recall that $\langle \cdot, \cdot \rangle$ denotes the inner product. Further, we use the shorthand notation $\mathsf{BRGap}_i(\boldsymbol{x}_i, \boldsymbol{u}_i) := \max_{\boldsymbol{x}_i' \in \Delta(\mathcal{A}_i)} \langle \boldsymbol{x}_i' - \boldsymbol{x}_i, \boldsymbol{u}_i \rangle$ for the best-response gap.

The predominant solution concept in game theory is the *Nash equilibrium* (Nash, 1950).

**Definition 2.1.** A strategy profile $(\boldsymbol{x}_1, \ldots, \boldsymbol{x}_n) \in \Delta(\mathcal{A}_1) \times \cdots \times \Delta(\mathcal{A}_n)$ is an $\epsilon$-*Nash equilibrium* if for any player $i \in [n]$ and unilateral deviation $\boldsymbol{x}_i' \in \Delta(\mathcal{A}_i)$, $u_i(\boldsymbol{x}_i', \boldsymbol{x}_{-i}) \leq u_i(\boldsymbol{x}_i, \boldsymbol{x}_{-i}) + \epsilon$.

A standard relaxation of the Nash equilibrium is the *coarse correlated equilibrium* (Definition B.1), which can be attained by no-regret algorithms (Proposition B.2). While finding a Nash equilibrium is hard even in two-player general-sum games (Daskalakis et al., 2008; Chen et al., 2009), our focus is on *potential games*—equivalently, *congestion games* (Monderer & Shapley, 1996).

**Potential games** This is a seminal class that goes back to the work of Rosenthal (1973). The defining property is the admission of a global, player-independent function—the *potential*—whose difference reflects the benefit of any unilateral deviation.

**Definition 2.2** (Potential game). An $n$-player game is a *potential game* if there exists a function $\Phi : \mathcal{A}_1 \times \cdots \times \mathcal{A}_n \to \mathbb{R}$ such that for any player $i \in [n]$ and actions $a_i, a_i' \in \mathcal{A}_i$, $\boldsymbol{a}_{-i} \in \bigtimes_{i' \neq i} \mathcal{A}_{i'}$,

$$\Phi(a_i', \boldsymbol{a}_{-i}) - \Phi(a_i, \boldsymbol{a}_{-i}) = u_i(a_i', \boldsymbol{a}_{-i}) - u_i(a_i, \boldsymbol{a}_{-i}). \tag{1}$$

A special case of a potential game worth noting is an *identical-interest* game, which means that $u_1(\boldsymbol{x}_1, \ldots, \boldsymbol{x}_n) = \cdots = u_n(\boldsymbol{x}_1, \ldots, \boldsymbol{x}_n)$ for all $\boldsymbol{x}_1, \ldots, \boldsymbol{x}_n$. In the presence of only two players, this simplifies to $u_1(\boldsymbol{x}_1, \boldsymbol{x}_2) = \langle \boldsymbol{x}_1, \mathbf{A}\boldsymbol{x}_2 \rangle = u_2(\boldsymbol{x}_1, \boldsymbol{x}_2)$ for a common matrix $\mathbf{A} \in \mathbb{R}^{\mathcal{A}_1 \times \mathcal{A}_2}$.

A (mixed) Nash equilibrium in potential games is amenable to (projected) gradient descent, but is likely hard to compute when the precision $\epsilon > 0$ is exponentially small (Babichenko & Rubinstein, 2021). Our focus will be on algorithms whose complexity is polynomial in $1/\epsilon$.

**Constrained optimization** More broadly, beyond potential games, we are interested in computing *Karush-Kuhn-Tucker (KKT) points* of a function $u : \mathcal{X} \to \mathbb{R}$, where $\mathcal{X} := \Delta(\mathcal{A}_1) \times \cdots \times \Delta(\mathcal{A}_n)$. We assume that $u$, which is to be maximized, is differentiable over an open set $\hat{\mathcal{X}} \supset \mathcal{X}$ and $L$-*smooth*, meaning that $\|\nabla u(\boldsymbol{x}) - \nabla u(\boldsymbol{x}')\|_2 \leq L\|\boldsymbol{x} - \boldsymbol{x}'\|_2$ for all $\boldsymbol{x}, \boldsymbol{x}' \in \mathcal{X}$; we recall that $\|\boldsymbol{x}\|_2 := \sqrt{\langle \boldsymbol{x}, \boldsymbol{x} \rangle}$ denotes the (Euclidean) $\ell_2$ norm. We make the normalization assumption $|\langle \boldsymbol{x}_i - \boldsymbol{x}_i', \nabla_{\boldsymbol{x}_i} u(\boldsymbol{x}) \rangle| \leq 1$

for all $i \in [n]$ and $\boldsymbol{x}_i, \boldsymbol{x}'_i \in \Delta(\mathcal{A}_i)$. The goal is to minimize the *KKT gap*, which we measure by

$$\mathsf{KKTGap} : \mathcal{X} \ni \boldsymbol{x} \mapsto \max_{\boldsymbol{x}' \in \mathcal{X}} \langle \boldsymbol{x}' - \boldsymbol{x}, \nabla u(\boldsymbol{x}) \rangle = \sum_{i=1}^{n} \mathsf{BRGap}_i(\boldsymbol{x}_i, \nabla_{\boldsymbol{x}_i} u(\boldsymbol{x})). \qquad (2)$$

A point with small KKT gap per (2) is also referred to as an approximate *first-order stationary point*, which is an approximate fixed point of the (constrained) gradient descent mapping $\boldsymbol{x} \mapsto \Pi_{\mathcal{X}}(\boldsymbol{x} + \eta \nabla u(\boldsymbol{x}))$, where $\eta \leq 1/L$ and $\Pi_{\mathcal{X}}(\cdot)$ is (Euclidean) projection mapping. A potential game can be seen as the special case in which $u$ is multilinear.

## 2.2 ONLINE LEARNING AND REGRET MATCHING

Moving on, we now introduce $\mathsf{RM}$ and $\mathsf{RM}^+$ within the framework of online learning. Here, a *learner* interacts with an *environment* over a sequence of $T$ rounds. In each round $t \in [T]$, the learner first elects a mixed strategy $\boldsymbol{x} \in \Delta(\mathcal{A})$. The environment in turn specifies a linear utility function $u^{(t)} : \boldsymbol{x} \mapsto \langle \boldsymbol{x}, \boldsymbol{u}^{(t)} \rangle$ for some utility vector $\boldsymbol{u}^{(t)} \in \mathbb{R}^{\mathcal{A}}$; it is assumed that $u^{(t)}$ has a range bounded by 1. In the full-feedback setting, $\boldsymbol{u}^{(t)}$ is revealed to the learner at the end of the $t$th round. The performance of the learner in this online environment is evaluated through *regret*,

$$\mathsf{Reg}^{(T)} := \max_{\boldsymbol{x}' \in \Delta(\mathcal{A})} \sum_{t=1}^{T} \langle \boldsymbol{x}' - \boldsymbol{x}^{(t)}, \boldsymbol{u}^{(t)} \rangle. \qquad (3)$$

Two popular algorithms for minimizing regret on the simplex are regret matching ($\mathsf{RM}$) and regret matching$^+$ ($\mathsf{RM}^+$), formally defined in Algorithms 1 and 2. They both prescribe playing an action with probability proportional to the nonnegative regret accumulated by that action. Their *only* difference is that $\mathsf{RM}^+$ always truncates the regret to 0; $\mathbf{1}$ denotes the all-ones vector, whose dimension is omitted as it is clear from the context, and $[\cdot]^+ := \max(\mathbf{0}, \cdot)$ is the nonnegative part.

**Proposition 2.3** (Zinkevich et al., 2007; Farina et al., 2021). *For any sequence of utilities $(\boldsymbol{u}^{(t)})_{t=1}^{T}$, both $\mathsf{RM}$ and $\mathsf{RM}^+$ guarantee that the $\ell_2$ norm of $[\boldsymbol{r}^{(T)}]^+$ is at most $\sqrt{mT}$.*

In particular, for both $\mathsf{RM}$ and $\mathsf{RM}^+$, $\mathsf{Reg}^{(T)} \leq \|[\boldsymbol{r}^{(T)}]^+\|_\infty \leq \|[\boldsymbol{r}^{(T)}]^+\|_2 \leq \sqrt{mT}$.

---

**Algorithm 1:** Regret matching ($\mathsf{RM}$)

1 Initialize cumulative regrets $\boldsymbol{r}^{(0)} \leftarrow \mathbf{0}$;
2 Initialize strategy $\boldsymbol{x}^{(0)} \in \Delta(\mathcal{A})$;
3 **for** $t = 1, \ldots, T$ **do**
4      Set $\boldsymbol{\theta}^{(t)} \leftarrow [\boldsymbol{r}^{(t-1)}]^+$;
5      **if** $\boldsymbol{\theta}^{(t)} \neq \mathbf{0}$ **then**
6          Compute $\boldsymbol{x}^{(t)} \leftarrow \boldsymbol{\theta}^{(t)}/\|\boldsymbol{\theta}^{(t)}\|_1$;
7      **else**
8          $\boldsymbol{x}^{(t)} \leftarrow \boldsymbol{x}^{(t-1)}$;
9      Output strategy $\boldsymbol{x}^{(t)} \in \Delta(\mathcal{A})$ ;
10      Observe utility $\boldsymbol{u}^{(t)} \in \mathbb{R}^{\mathcal{A}}$;
11      $\boldsymbol{r}^{(t)} \leftarrow \boldsymbol{r}^{(t-1)} + \boldsymbol{u}^{(t)} - \langle \boldsymbol{x}^{(t)}, \boldsymbol{u}^{(t)} \rangle \mathbf{1}$;

**Algorithm 2:** Regret matching$^+$ ($\mathsf{RM}^+$)

1 Initialize cumulative regrets $\boldsymbol{r}^{(0)} := \mathbf{0}$;
2 Initialize strategy $\boldsymbol{x}^{(1)} \in \Delta(\mathcal{A})$;
3 **for** $t = 1, \ldots, T$ **do**
4      Set $\boldsymbol{\theta}^{(t)} \leftarrow \boldsymbol{r}^{(t-1)}$;
5      **if** $\boldsymbol{\theta}^{(t)} \neq \mathbf{0}$ **then**
6          Compute $\boldsymbol{x}^{(t)} \leftarrow \boldsymbol{\theta}^{(t)}/\|\boldsymbol{\theta}^{(t)}\|_1$;
7      **else**
8          $\boldsymbol{x}^{(t)} \leftarrow \boldsymbol{x}^{(t-1)}$;
9      Output strategy $\boldsymbol{x}^{(t)} \in \Delta(\mathcal{A})$ ;
10      Observe utility $\boldsymbol{u}^{(t)} \in \mathbb{R}^{\mathcal{A}}$;
11      $\boldsymbol{r}^{(t)} \leftarrow [\boldsymbol{r}^{(t-1)} + \boldsymbol{u}^{(t)} - \langle \boldsymbol{x}^{(t)}, \boldsymbol{u}^{(t)} \rangle \mathbf{1}]^+$;

---

**Simultaneous and alternating updates** We are interested in the convergence of $\mathsf{RM}$ and $\mathsf{RM}^+$ when used by all players; in the constrained optimization setting, we think of having one player acting on each simplex, in direct correspondence with potential games. In this setting, the sequence of utilities $(\boldsymbol{u}_i^{(t)})_{t=1}^{T}$ given as input to player $i \in [n]$ is determined by the strategies of the other players. If the updates are *simultaneous*, we have $\boldsymbol{u}_i^{(t)} = \nabla_{\boldsymbol{x}_i} u(\boldsymbol{x}^{(t)})$ for each player $i \in [n]$. (In potential games, the potential function $\Phi$ plays the role of $u$, so that $\boldsymbol{u}_i^{(t)} = \boldsymbol{u}_i(\boldsymbol{x}_{-i}^{(t)})$.) In the alternating setting, we go through the players in a round-robin fashion $i = 1, \ldots, n$. In the *lazy* version of the update, for a fixed precision $\epsilon > 0$, we first compute $\boldsymbol{u}_i^{(t)} = \nabla_{\boldsymbol{x}_i} u(\boldsymbol{x}_{i'<i}^{(t+1)}, \boldsymbol{x}_{i' \geq i}^{(t)})$. If the best-response gap of player $i \in [n]$ is already at most $\epsilon$, we refrain from updating that player,

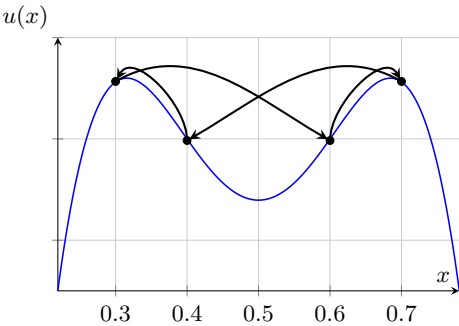

Figure 2: The example corresponding to Proposition 3.2, demonstrating that having zero regret, let alone sublinear, has no implications concerning convergence in terms of KKT gap.

so that $\boldsymbol{x}_i^{(t+1)} \coloneqq \boldsymbol{x}_i^{(t)}$. Otherwise, the player updates its strategy to $\boldsymbol{x}_i^{(t+1)}$ using $\boldsymbol{u}_i^{(t)}$. We refer to this scheme as $\epsilon$-*lazy alternating* updates ($\epsilon$-lazy simultaneous updates are defined similarly); one limitation of this lazy variant is that it is not an "anytime algorithm," in that one needs to specify the precision beforehand. In the more common, non-lazy version of alternation, a player is updated regardless of the best-response gap; in the sequel, we obtain convergence results for both variants.

## 3 CONVERGENCE OF REGRET MATCHING$^+$

In this section, we analyze the convergence of RM$^+$ in potential games (Section 3.1), and more broadly, constrained optimization (Section 3.2). A central theme in our analysis of RM and RM$^+$ is a recurring connection between regret and convergence to KKT points.

Before we proceed, it is worth highlighting that, in general, the no-regret property is fundamentally different from convergence to KKT points in *nonconvex problems*. To begin with, we point out that when the underlying function to be maximized, $u$, is concave, then the no-regret property does imply convergence to a global optimum, from Jensen's inequality.

**Proposition 3.1** (Under concavity, no-regret implies convergence)**.** *Let $u$ be a smooth concave function. If an online algorithm observes the sequence of utilities $(\nabla u(\boldsymbol{x}^{(t)}))_{t=1}^T$, then $\frac{1}{T} \sum_{t=1}^T u(\boldsymbol{x}^{(t)}) \geq \max_{\boldsymbol{x} \in \mathcal{X}} u(\boldsymbol{x}) - \frac{1}{T}\mathsf{Reg}^{(T)}$, where $\mathsf{Reg}^{(T)}$ is the regret of the algorithm per* (3).

Thus, if the algorithm has vanishing average regret, $u(\boldsymbol{x}^{(t)})$ converges to $\max_{\boldsymbol{x} \in \mathcal{X}} u(\boldsymbol{x})$ (in density). But beyond concave problems, no-regret algorithms do not necessarily guarantee convergence even to a KKT point, as we point out below.

**Proposition 3.2.** *For any $T \in \mathbb{N}$ with $T = 0 \mod 4$, there exists a polynomial function $u$ in $[0, 1]$ and a sequence of points $(x^{(t)})_{t=1}^T$ such that*

- *the regret of the sequence with respect to $(\nabla u(x^{(t)}))_{t=1}^T$ is zero, while*
- *every point in the sequence has an $\Omega(1)$ KKT gap with respect to the function $u$.*

This is based on the 4-cycle $0.6 \to 0.7 \to 0.4 \to 0.3 \to 0.6$. If the gradients observed at those points are $0.6 \mapsto 2, 0.7 \mapsto -1, 0.4 \mapsto -2, 0.3 \mapsto 1$, it follows that i) $\sum_{t=1}^T \nabla u(x^{(t)}) = 0$ and ii) $\sum_{t=1}^T x^{(t)} \nabla u(x^{(t)}) = 0$, which in turn implies that this sequence incurs zero regret. But, by construction, the gradients at those interior points have a large magnitude, which in turn implies that the KKT gap is large. (That the average is a local minimum is coincidental.) A polynomial consistent with the above gradients is $90x - 298.\bar{3}x^2 + 416.\bar{6}x^3 - 208.\bar{3}x^4$, leading to Proposition 3.2. We note that the above sequence of iterates is not realizable through an algorithm such as gradient descent.

### 3.1 POTENTIAL GAMES

We first analyze convergence in potential games. A key property, which paves the way for Theorem 3.4, is that, for a fixed utility vector, RM$^+$ has a one-step improvement property; the lemma below takes the perspective of a single, arbitrary player in the game.

**Lemma 3.3** (One-step improvement of $\text{RM}^+$). *For any $r \in \mathbb{R}_{\geq 0}^{\mathcal{A}}$ and $u \in \mathbb{R}^{\mathcal{A}}$, we define $x := r/\|r\|_1$; if $r = 0$, $x \in \Delta(\mathcal{A})$ can be arbitrary. If $r' := [r + u - \langle x, u \rangle \mathbf{1}]^+ \neq 0$ and $x' := r'/\|r'\|_1$, then*

$$\langle x' - x, u \rangle \geq \frac{1}{\|r'\|_1} \left( \max_{a \in \mathcal{A}} u[a] - \langle x, u \rangle \right)^2 = \frac{1}{\|r'\|_1} \text{BRGap}(x, u)^2. \tag{4}$$

*If $r' = 0$, then $\langle x, u \rangle = \langle x', u \rangle \geq \max_{a \in \mathcal{A}} u[a]$.*

The left-hand side of (4) reflects the improvement in utility obtained by updating $x$ to $x'$; in particular, $\max_{a \in \mathcal{A}} u[a] - \langle x, u \rangle$ is the best-response gap of $x$ with respect to $u$. Lemma 3.3 implies that the utility is monotonically increasing—unless the current strategy is already a best response to $u$. Furthermore, so long as the regret vector is *small enough*, the improvement is bound to be substantial, being proportional to the squared best-response gap. It is worth noting that Lemma 3.3 holds no matter the initial regret vector $r$, subject to $r \in \mathbb{R}_{\geq 0}$; this invariance always holds for $\text{RM}^+$ (by definition of the update in Algorithm 2), but that is not so for RM (*cf.* Lemma 3.6).

**Convergence in potential games** We now employ Lemma 3.3 to show that alternating $\text{RM}^+$ quickly converges to approximate Nash equilibria in potential games. Using the fact that the game admits a potential (per Definition 2.2), we have that for any round $t \in [T]$, $\Phi(x_1^{(t+1)}, \ldots, x_n^{(t+1)}) - \Phi(x_1^{(t)}, \ldots, x_n^{(t)}) \geq \sum_{i=1}^n \frac{1}{\|r_i^{(t)}\|_1} \text{BRGap}_i(x_i^{(t)}, u_i^{(t)})^2 \mathbb{1}\{\text{BRGap}_i(x_i^{(t)}, u_i^{(t)}) > \epsilon\}$, where we used Lemma 3.3 together with the assumption that only players with more than $\epsilon$ best-response gap update their strategies. The telescopic summation over $t = 1, \ldots, T$ yields

$$\Phi_{\text{range}} \geq \sum_{t=1}^T \sum_{i=1}^n \frac{1}{\|r_i^{(t)}\|_1} \text{BRGap}_i(x_i^{(t)}, u_i^{(t)})^2 \mathbb{1}\{\text{BRGap}_i(x_i^{(t)}, u_i^{(t)}) > \epsilon\}, \tag{5}$$

where $\Phi_{\text{range}}$ denotes the range of the potential function. If in every round $t \in [T]$ there is a player $i \in [n]$ such that $\text{BRGap}_i(x_i^{(t)}, u_i^{(t)}) > \epsilon$, we have $\Phi_{\text{range}} \geq \sum_{t=1}^T \frac{1}{m\sqrt{t}} \epsilon^2 \geq \frac{1}{m} \epsilon^2 \sqrt{T}$, where we used that $\|r_i^{(t)}\|_1 \leq \sqrt{m}\|r_i^{(t)}\|_2 \leq m\sqrt{T}$ (Proposition 2.3). We thus arrive at the following result.

**Theorem 3.4.** *In any potential game, $\epsilon$-lazy alternating $\text{RM}^+$ requires at most $1 + \frac{(m\Phi_{\text{range}})^2}{\epsilon^4}$ rounds to converge to an $\epsilon$-Nash equilibrium. More broadly, if $\|r_i^{(t)}\|_1 \leq C(n, m)t^\alpha$ for all $i \in [n]$ and some $\alpha \in [0, 1/2]$, it requires $1 + \frac{(C(n,m)\Phi_{\text{range}})^\beta}{\epsilon^{2\beta}}$ rounds, where $\beta := 1/1-\alpha$.*

This provides a convergence rate of $T^{-1/4}$ to Nash equilibria. Notwithstanding Proposition 3.2, an intriguing aspect of Theorem 3.4 is that it *connects convergence to Nash equilibria to the regret* of $\text{RM}^+$. In particular, if $\text{RM}^+$ did not have the no-regret property, meaning that $\|r_i^{(t)}\|_1 = \Omega(t)$, one could only prove an exponential bound since $\sum_{t=1}^T 1/t = \Theta(\log T)$. At the other end, when each player accumulates constant regret, Theorem 3.4 implies an improved convergence rate of $T^{-1/2}$.

One caveat of lazy alternating $\text{RM}^+$ prescribed by Theorem 3.4 is that the desired precision should be known in advance in order to execute the algorithm. We address this limitation in Theorem C.7, where we show that the usual, non-lazy version also converges after $O_\epsilon(1/\epsilon^4)$ rounds, albeit at the cost of introducing an additional dependence in the total number of iterations.

**Faster rates using discounting** Next, we refine Theorem 3.4 through the use of *discounted* $\text{RM}^+$, which means that the regret vector is multiplied by a discount factor $\alpha^{(t)} \in (0, 1]$ in each round; we spell out $\text{DRM}^+$ in Algorithm 3. This class of algorithms was introduced by Brown & Sandholm (2019a), who showed that discounting drastically improves empirical performance in zero-sum games. Our next result shows that $\text{DRM}^+$ with geometric discounting, that is, $\alpha^{(t)} = 1 - \gamma$ for some time-invariant $\gamma \in (0, 1)$, attains a rate of $T^{-1/2}$ to Nash equilibria in potential games; this is considerably faster than the $T^{-1/4}$ rate for alternating $\text{RM}^+$ guaranteed by Theorem 3.4. The basic reason is that $\text{DRM}^+$—with geometric discounting—maintains the norm of the regret vector bounded by $\sqrt{m/\gamma}$ (Lemma C.2 and Corollary C.3), while still enjoying the one-step improvement property.

**Corollary 3.5.** *In any potential game, $\epsilon$-lazy alternating $\text{DRM}^+$ with discount factor $1 - \gamma \in (0, 1)$ requires at most $1 + \frac{m\Phi_{\text{range}}}{\epsilon^2 \sqrt{\gamma}}$ rounds to converge to an $\epsilon$-Nash equilibrium.*

To our knowledge, this is the first time that discounting yields a provable, worst-case improvement over the bound obtained for non-discounted $\texttt{RM}^+$.

**Regret matching**   Before we switch gears to the more general constrained optimization setting, it is instructive to examine the behavior of $\texttt{RM}$. It turns out that one can adjust Lemma 3.3, but with a crucial caveat: the one-step improvement property is now only *conditional*, as specified below.

**Lemma 3.6.** *For any $\boldsymbol{r} \in \mathbb{R}_{\geq 0}^{\mathcal{A}}$ and $\boldsymbol{u} \in \mathbb{R}^{\mathcal{A}}$, we define $\boldsymbol{x} := \boldsymbol{\theta}/\|\boldsymbol{\theta}\|_1$, where $\boldsymbol{\theta} := \max(\boldsymbol{r}, \mathbf{0})$; if $\boldsymbol{\theta} = \mathbf{0}$, $\boldsymbol{x} \in \Delta(\mathcal{A})$ can be arbitrary. If $\boldsymbol{r}' := \boldsymbol{r} + \boldsymbol{u} - \langle \boldsymbol{x}, \boldsymbol{u} \rangle \mathbf{1}$ and $\boldsymbol{x}' := \boldsymbol{\theta}'/\|\boldsymbol{\theta}'\|_1$, where $\boldsymbol{\theta}' = \max(\boldsymbol{r}', \mathbf{0}) \neq \mathbf{0}$, then $\langle \boldsymbol{x}' - \boldsymbol{x}, \boldsymbol{u} \rangle \geq \frac{1}{\|\boldsymbol{\theta}'\|_1} \|\boldsymbol{\theta}' - \boldsymbol{\theta}\|_2^2 \geq \frac{1}{\|\boldsymbol{\theta}'\|_1} (\max_{a \in \mathcal{A}} \boldsymbol{u}[a] - \langle \boldsymbol{x}, \boldsymbol{u} \rangle)^2 \mathbb{1}\{\boldsymbol{r}[a] \geq 0\}$, where $a \in \arg\max_{a' \in \mathcal{A}} \boldsymbol{u}[a']$. If $\boldsymbol{\theta}' = \mathbf{0}$, then $\langle \boldsymbol{x}, \boldsymbol{u} \rangle = \langle \boldsymbol{x}', \boldsymbol{u} \rangle \geq \boldsymbol{u}[a]$.*

We see that $\texttt{RM}$'s one-step improvement is conditional on the regret accumulated thus far by a best-response action to be nonnegative. This is not an artifact of our analysis; it alludes to a fundamental discrepancy between $\texttt{RM}$ and $\texttt{RM}^+$ that will be formally established later on (Theorem 4.4). The main issue with $\texttt{RM}$ can be seen as follows. If we consider a utility vector $\boldsymbol{u} = (1, 0)$ and the initial regret vector is, say, $(-R, R)$, it will take $\texttt{RM}$ many iterations—proportionally to the magnitude of $R > 0$—to finally change strategies, although this will eventually happen with a stationary utility.

## 3.2   Constrained optimization and simultaneous updates

We now treat the more general setting where we are maximizing an $L$-smooth function $u$.

**Single simplex**   We begin with the special case of a single probability simplex, $\mathcal{X} = \Delta(\mathcal{A})$. Our first goal is to adapt Lemma 3.3. The key challenge is that $\texttt{RM}^+$ does *not* have a one-step improvement, unlike algorithms such as gradient descent (for a small enough learning rate), even if one initializes $\texttt{RM}^+$ close to a KKT point. But we observe that if the norm of the regret vector is *large enough*—having small regrets is an obstacle here, in contrast to Section 3.1—we are guaranteed a one-step improvement in terms of the value of the function (Lemma 3.7).

To do so, we will use the basic quadratic bound, which yields $u(\boldsymbol{x}') \geq u(\boldsymbol{x}) + \langle \nabla u(\boldsymbol{x}), \boldsymbol{x}' - \boldsymbol{x} \rangle - \frac{L}{2}\|\boldsymbol{x} - \boldsymbol{x}'\|_2^2$; we think of $\boldsymbol{x}'$ as the updated strategy starting from $\boldsymbol{x}$. Using a slight refinement of Lemma 3.3, we first have the lower bound $\langle \boldsymbol{x}' - \boldsymbol{x}, \nabla u(\boldsymbol{x}) \rangle \geq \frac{1}{\|\boldsymbol{r}'\|_1} \|\boldsymbol{r} - \boldsymbol{r}'\|_2^2$ (Lemma C.5).

Also, we observe that $\|\boldsymbol{x} - \boldsymbol{x}'\|_1 \leq \|\boldsymbol{r} - \boldsymbol{r}'\|_1 \left( \frac{1}{\|\boldsymbol{r}\|_1} + \frac{1}{\|\boldsymbol{r}'\|_1} \right)$ (Lemma C.6). We are now ready to establish a *conditional* one-step improvement when the regret vector has a *sufficiently large* norm.

**Lemma 3.7.** *Let $u$ be an $L$-smooth function over $\Delta(\mathcal{A})$. For any $\boldsymbol{r} \in \mathbb{R}_{\geq 0}^{\mathcal{A}}$ with $\boldsymbol{r} \neq \mathbf{0}$, we define $\boldsymbol{x} := \boldsymbol{r}/\|\boldsymbol{r}\|_1$. Further, let $\boldsymbol{r}' := [\boldsymbol{r} + \nabla u(\boldsymbol{x}) - \langle \boldsymbol{x}, \nabla u(\boldsymbol{x}) \rangle \mathbf{1}]^+ \neq \mathbf{0}$ and $\boldsymbol{x}' := \boldsymbol{r}'/\|\boldsymbol{r}'\|_1$. If $\|\boldsymbol{r}'\|_2 \geq \max\{2m, 9mL\}$, then $u(\boldsymbol{x}') - u(\boldsymbol{x}) \geq \frac{1}{2\|\boldsymbol{r}'\|_1} (\max_{\boldsymbol{x}^\star \in \Delta(\mathcal{A})} \langle \boldsymbol{x}^\star - \boldsymbol{x}, \nabla u(\boldsymbol{x}) \rangle)^2$.*

Lemma 3.7 only shows a one-step improvement so long as the norm of the regret vector is large enough. But how can we guarantee that? It would seem possible that $\texttt{RM}^+$ ends up cycling in perpetuity under a regret vector with small norm. The following lemma shows that cannot happen; it turns out that maintaining this monotonicity property for the norm of the regret vector is crucial for designing an optimal variant of $\texttt{RM}^+$ in zero-sum games (Zhang et al., 2025).

**Lemma 3.8.** *For any $t$, $\texttt{RM}^+$ guarantees $\|\boldsymbol{r}^{(t)}\|_2^2 \geq \|\boldsymbol{r}^{(t-1)}\|_2^2 + \|[\boldsymbol{g}^{(t)}]^+\|_2^2$, where $\boldsymbol{g}^{(t)} := \nabla u(\boldsymbol{x}^{(t)}) - \langle \nabla u(\boldsymbol{x}^{(t)}), \boldsymbol{x}^{(t)} \rangle \mathbf{1}$ is the instantaneous regret vector at round $t$.*

In particular,

$$\|\boldsymbol{r}^{(t)}\|_2^2 \geq \|\boldsymbol{r}^{(t-1)}\|_2^2 + \|[\boldsymbol{g}^{(t)}]_+\|_2^2 \geq \|\boldsymbol{r}^{(t-1)}\|_2^2 + \mathsf{KKTGap}(\boldsymbol{x}^{(t)})^2$$

since $\|[\boldsymbol{g}^{(t)}]^+\|_2^2 \geq \mathsf{KKTGap}(\boldsymbol{x}^{(t)})^2$. That is, not only is the $\ell_2$ norm of the regret vector nondecreasing, but the increase is at least $\mathsf{KKTGap}(\boldsymbol{x}^{(t)})^2$ at each round $t \in [T]$. Combining with Lemma 3.7 yields the following.

**Theorem 3.9.** *Let $u$ be an $L$-smooth function in $\Delta(\mathcal{A}) \subset \mathbb{R}^m$ with range $u_{\text{range}}$ and $R := \max\{2m, 9mL\}$. $\texttt{RM}^+$ requires at most $1 + \frac{(m(2u_{\text{range}} + R^2))^2}{\epsilon^4}$ rounds to reach an $\epsilon$-KKT point.*

**Simultaneous updates in symmetric potential games**   We now use Theorem 3.9 to prove convergence of *simultaneous* RM$^+$ in *symmetric* potential games; our earlier result in Theorem 3.4 shows convergence for arbitrary potential games but for the alternating version. The symmetry assumption here means that $\mathcal{A}_1 = \mathcal{A}_1 = \cdots = \mathcal{A}_n = \mathcal{A}$ and $\boldsymbol{u}_1(\boldsymbol{x}_{-1}) = \boldsymbol{u}_2(\boldsymbol{x}_{-2}) = \cdots = \boldsymbol{u}_n(\boldsymbol{x}_{-n})$ when $\boldsymbol{x}_1 = \boldsymbol{x}_2 = \cdots = \boldsymbol{x}_n$. It is further assumed that all players initialize from the same strategy, so that the previous property implies that, inductively, it will be the case that $\boldsymbol{x}_1^{(t)} = \boldsymbol{x}_2^{(t)} = \cdots = \boldsymbol{x}_n^{(t)}$ for all $t$ under simultaneous updates because players observe exactly the same utility vector. A simple example of this is a two-player game with a common, symmetric payoff matrix $\mathbf{A} = \mathbf{A}^\top$. Then $\boldsymbol{u}_1(\boldsymbol{x}_2) = \mathbf{A}\boldsymbol{x}_2$ and $\boldsymbol{u}_2(\boldsymbol{x}_1) = \mathbf{A}\boldsymbol{x}_1$, so the previous assumption is satisfied.

**Corollary 3.10.** *In any symmetric potential game, simultaneous RM$^+$ converges to an $\epsilon$-Nash equilibrium after $O_\epsilon(1/\epsilon^4)$ rounds. In particular, if convergence to the set of CCE happens at a rate of $T^{-(1-\alpha)}$, for some $\alpha \in [0, 1/2]$, the rate of convergence to Nash equilibria is no worse than $T^{-\frac{1-\alpha}{2}}$.*

**Multiple simplices**   We now have the necessary tools to analyze the general case where we maximize $u$ over a product of simplices. Similarly to Theorem 3.4, we run alternating RM$^+$, thinking of every individual simplex as being controlled by a single player; this is akin to coordinate descent.

**Corollary 3.11.** *If $u$ is an $L$-smooth function in $\Delta(\mathcal{A}_1) \times \cdots \times \Delta(\mathcal{A}_n)$ with range $u_{range}$, $\epsilon$-lazy alternating RM$^+$ initialized at $\boldsymbol{r}_i^{(0)} = \max\{2\sqrt{m_i}, 9\sqrt{m_i}L\}\mathbf{1}$ for each player $i \in [n]$ requires at most $1 + \frac{4n^4m^2u_{range}^2}{\epsilon^4}$ rounds to reach an $\epsilon$-KKT point of $u$.*

The proof follows directly from Lemma 3.7 together with a telescopic summation. The non-lazy version of RM$^+$ admits a qualitatively similar bound, following Theorem C.7. Furthermore, a similar bound holds even under simultaneous RM$^+$ (Corollary C.11), which follows by extending Lemma 3.7 to multiple simplices (Lemma C.10).

One caveat of those results is that the regret vector of each player needs to be initialized at a specific threshold. Our next result addresses this limitation by analyzing the usual parameter-free and scale-invariant version of RM$^+$, at the cost of introducing a worse dependence on $1/\epsilon$.

**Theorem 3.12.** *If $u$ is an $L$-smooth function in $\Delta(\mathcal{A}_1) \times \cdots \times \Delta(\mathcal{A}_n)$ with range $u_{range}$, $\epsilon$-lazy alternating (or simultaneous) RM$^+$ requires at most $O_\epsilon(1/\epsilon^8)$ rounds to reach an $\epsilon$-KKT point of $u$.*

## 4   EXPONENTIAL LOWER BOUNDS FOR REGRET MATCHING

In stark contrast, we show that RM, with or without alternation, can take exponentially many rounds to reach an approximate Nash equilibrium even in two-player identical-interest games. The underlying class of games is based on the one considered by Panageas et al. (2023a), who treated fictitious play. Specifically, for $m = 4, 6, \ldots$ and $k \in \mathbb{N}$ we define the matrix $\mathbf{A}_{m,k}$ per the recursion

$$\mathbb{R}^{m \times m} \ni \mathbf{A}_{m,k} := \begin{bmatrix} k+1 & 0 & \cdots & 0 & 0 \\ 0 & & & & k+4 \\ \vdots & & \mathbf{A}_{m-2,k+4} & & \vdots \\ 0 & & & & 0 \\ k+2 & 0 & \cdots & 0 & k+3 \end{bmatrix}, \text{ where } \mathbf{A}_{2,k} := \begin{bmatrix} k+1 & 0 \\ k+2 & k+3 \end{bmatrix}.$$

(An illustrative example appears in Section C.2.) For any even dimension $m$, we define $\mathbf{A} := \mathbf{A}_{m,0}$, with maximum entry $2m - 1$. Further, we define, for $1 \le a_1 \le m + 1$ and $1 \le a_2 \le m + 1$,

$$\mathbf{B}[a_1, a_2] := \begin{cases} \mathbf{A}[a_1, a_2] & \text{if } a_1 \le m \text{ and } a_2 \le m; \\ 1/2 & \text{if } (a_1 = m+1 \text{ and } a_2 = 1) \text{ or } (a_1 = 1 \text{ and } a_2 = m+1); \\ 0 & \text{otherwise.} \end{cases} \quad (6)$$

The action sets of the two players are $\mathcal{A}_1 = [m + 1] = \mathcal{A}_2$. We assume that RM is initialized at the pure strategy $(m + 1, m + 1)$; Section C.2 shows how to adapt the lower bound when RM is initialized at the uniform random strategy (Corollary C.16), which is more common. We recall that one round includes one update from each player, which for now is assumed to be made in a simultaneous fashion. For a payoff $k \in \mathbb{N}$, we denote by $a_1(k), a_2(k) \in [m]$ the row and column index, respectively, corresponding to $k$ in the matrix $\mathbf{A}$.

We begin by stating a basic invariance concerning the behavior of RM when executed on the game (6).

**Property 4.1.** *After the first round both players play the first action. Thereupon, either the players play with probability $1$ $(a_1(k), a_2(k))$, or, when $k$ is odd, only Player 1 (respectively, Player 2 when $k$ is even) mixes between $a_1(k)$ and $a_1(k+1)$ (respectively, $a_2(k)$ and $a_2(k+1)$). If a row or a column stops being played, it will never be played henceforth. An action profile $(a_1(k+1), a_2(k+1))$ is played with positive probability only if $(a_1(k), a_2(k))$ was played at some previous round.*

We prove this property inductively in Section C.2. We take it for granted in what follows.

In accordance with Property 4.1, for $k \geq 2$, we define $\underline{t_k}$ to be the first round in which the action profile corresponding to payoff $k$ is played with positive probability and $\overline{t_k}$ the last round before the action profile corresponding to payoff $k+1$ is played with positive probability. We then define $T_k := \overline{t_k} - \underline{t_k} + 1$ to be the number of rounds corresponding to the period $[\underline{t_k}, \overline{t_k}]$.

We also define $\mathcal{A}_1(k) := \{a_1(k') : 2m - 1 \geq k' \geq k\}$ and $\mathcal{A}_2(k) := \{a_2(k') : 2m - 1 \geq k' \geq k\}$. These are the rows and columns, respectively, that will be played after the action profile corresponding to $k$ starts being played. The next crucial lemma shows that before an action becomes desirable, it will have accumulated very negative regret in the previous rounds.

**Lemma 4.2.** *For any even $k \geq 4$, let $\boldsymbol{r}_1^{(\overline{t_k}-2)}[a_1]$ be the regret of Player 1 with respect to any action $a_1 \in \mathcal{A}_1(k)$. Then $\boldsymbol{r}_1^{(\overline{t_k}-2)}[a_1] \leq -\sum_{l=2}^{k-2}(l-1)T_l$. Similarly, for any odd $k \geq 5$, if $\boldsymbol{r}_2^{(\overline{t_k}-2)}[a_2]$ is the regret of Player 2 with respect to any action $a_2 \in \mathcal{A}_2(k)$, $\boldsymbol{r}_2^{(\overline{t_k}-2)}[a_2] \leq -\sum_{l=2}^{k-2}(l-1)T_l$.*

At the same time, when an action has very negative regret, it will take a long time before that action gets played with positive probability, as formalized below.

**Lemma 4.3.** *For any even $k \geq 4$, $T_k \geq -\frac{1}{2}\boldsymbol{r}_2^{(\overline{t_k}-1)}[a_2(k+1)]$. Similarly, for every odd $k \geq 5$, $T_k \geq -\frac{1}{2}\boldsymbol{r}_1^{(\overline{t_k}-1)}[a_1(k+1)]$.*

By Lemmas 4.2 and 4.3, it follows that $T_k \geq \sum_{l=2}^{k-1}\frac{l-1}{2}T_l$ for any $k \geq 4$. By the inductive basis, we know that $T_3 \geq 1$. As a result, $T_k \geq \frac{k-2}{2}T_{k-1} \geq \frac{k-2}{2}\frac{k-3}{2}\ldots\frac{2}{2}T_3 \geq \frac{(k-2)!}{2^{k-3}}$ for all $k \geq 4$.

Moreover, it takes as least $T_{2m-2}$ rounds to converge to an NE with approximation gap at most $1/2m+2$ (Lemma C.15). We thus arrive at the following exponential lower bound.

**Theorem 4.4.** *Simultaneous RM requires $m^{\Omega(m)}$ rounds to converge to a $\frac{1}{2m}$-Nash equilibrium in two-player $m \times m$ identical-interest games.*

The same reasoning directly applies to alternating RM.

**Corollary 4.5.** *Alternating RM requires $m^{\Omega(m)}$ rounds to converge to a $\frac{1}{2m}$-Nash equilibrium in two-player $m \times m$ identical-interest games.*

## 5 FUTURE RESEARCH

Our paper sheds new light on the convergence properties of regret matching($^+$) in constrained optimization problems in general, and potential games in particular. We showed that RM$^+$ is a sound and fast first-order optimizer; on the flip side, RM can be exponentially slow even in two-player identical-interest games. Several interesting questions remain open. It would be interesting to understand whether RM$^+$, with or without alternation, can experience $\Omega(\sqrt{T})$ regret in potential games; this is known to be the case in zero-sum games (Farina et al., 2023), but remains unclear for the class of potential games. In light of our results, any improvement over the $\sqrt{T}$ barrier would automatically translate into a faster convergence rate to Nash equilibria. Moreover, we have worked exclusively in the full feedback setting; extending our results under stochastic or bandit feedback would be a natural next step. Finally, does RM asymptotically converge even under alternating updates?

## ACKNOWLEDGMENTS

Emanuel Tewolde and Vincent Conitzer thank the Cooperative AI Foundation, Macroscopic Ventures and Jaan Tallinn's donor-advised fund at Founders Pledge for financial support. Emanuel

Tewolde is also supported in part by the Cooperative AI PhD Fellowship. Ioannis Panageas is supported by NSF grant CCF-2454115. Tuomas Sandholm is supported by the Vannevar Bush Faculty Fellowship ONR N00014-23-1-2876, National Science Foundation grants RI-2312342 and RI-1901403, ARO award W911NF2210266, and NIH award A240108S001.

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

## A    FURTHER RELATED WORK

Much of the existing research on regret matching revolves around zero-sum games. Many variants have been proposed over the years to speed up its convergence (Xu et al., 2024b; Cai et al., 2025; Chakrabarti et al., 2024; Meng et al., 2025; Farina et al., 2021; Tammelin, 2014; Brown & Sandholm, 2019a). Some notable variations that have considerably improved performance are *predictive* RM and RM$^+$ (Farina et al., 2021), which rely on predicting the next utility, and *discounted* RM and RM$^+$ (Brown & Sandholm, 2019a; Zhang et al., 2024; Xu et al., 2024a), where one dynamically discounts the accumulated regret; in a similar vein, our work shows that a discounted variant of RM$^+$ achieves a better convergence upper bound than RM$^+$ in our setting (Corollary 3.5). Moreover, alternation is known to speed up performance, at least in zero-sum games (Tammelin, 2014), and has been the subject of much recent research (Wibisono et al., 2022; Cevher et al., 2023; Nan et al., 2025). It must be stressed that the focus of all that prior work was on zero-sum games. Constrained optimization is a fundamentally different problem. For one, in zero-sum games, it is only the average strategy of RM and RM$^+$ that converges, not the last iterate (Farina et al., 2023).

The recent paper of Tewolde et al. (2025) demonstrated that the regret matching family is a formidable first-order optimizer in constrained optimization problems. In particular, their focus was on (single-player) imperfect-recall problems, which are tantamount to general polynomial optimization problems over a product of simplices. Interestingly, many of the trends observed in zero-sum games are actually reversed in constrained optimization. For example, the predictive versions of RM and RM$^+$ generally performed worse than their non-predictive counterparts. One trend that did persist was the superiority of RM$^+$ over RM. It is also worth mentioning an earlier work by Ma & Gerber (2014) that also reported fast empirical convergence in a certain class of congestion games. Yet, there was hitherto no theoretical understanding of those algorithms in this setting. The main precursors of our work are the paper of Hart & Mas-Colell (2003), which established asymptotic convergence in discrete time but for a somewhat artificial variant of regret matching, and the paper of Marden et al. (2007), which analyzed asymptotically a certain variant of regret matching that aggressively discounts the regrets (*cf.* Corollary 3.5).

An interesting result that sheds light on RM and RM$^+$ is by Farina et al. (2021), who showed that RM can be obtained by running *follow the regularized leader (FTRL)* in a certain lifted space, whereas RM$^+$ can be obtained through *mirror descent (MD)* in the same space; this is despite the fact that, unlike FTRL and MD, RM and RM$^+$ are both parameter free. On a related note, Cai et al. (2024) showed that only forgetful algorithms—closer to MD than to FTRL—can attain fast last-iterate convergence. Our exponential separation of RM and RM$^+$ echoes their finding, although in a different setting and class of algorithms.

Zooming out of the RM family, understanding the convergence of no-regret dynamics in potential games has been a popular research topic (Kleinberg et al., 2009; Héliou et al., 2017; Palaiopanos et al., 2017; Panageas et al., 2023b; Cui et al., 2022; Blum et al., 2006). Our research also relates to parameter-free optimization; for example, we refer to Ivgi et al. (2023); Orabona & Pál (2016); Defazio & Mishchenko (2023) and references therein.

## B    FURTHER BACKGROUND

**Coarse correlated equilibria**    For completeness, we provide the definition of a coarse correlated equilibrium (Moulin & Vial, 1978), which is a relaxation of correlated equilibria (Aumann, 1974). The key connection that relates to our results is that if all players in a normal-form game have sublinear regret, the average correlated distribution of play converges to the set of coarse correlated equilibria. In particular, the rate of convergence is driven by the maximum of the players' regrets (Proposition B.2).

**Definition B.1** (Coarse correlated equilibrium). Consider an $n$-player game in normal form. A correlated distribution $\mu \in \Delta(\mathcal{A}_1 \times \cdots \times \mathcal{A}_n)$ is an $\epsilon$-*coarse correlated equilibrium (CCE)* if for any player $i \in [n]$ and deviation $a_i' \in \mathcal{A}_i$,

$$\mathbb{E}_{(a_1,\ldots,a_n)\sim\mu} u_i(a_1,\ldots,a_n) \geq \mathbb{E}_{(a_1,\ldots,a_n)\sim\mu} u_i(a_i', a_{-i}) - \epsilon.$$

**Proposition B.2.** *If each player $i \in [n]$ observes the sequence of utilities $(\boldsymbol{u}_i(\boldsymbol{x}_{-i}^{(t)}))_{t=1}^T$, the average correlated distribution of play is an $\epsilon$-CCE with $\epsilon \leq \frac{1}{T} \max_{1 \leq i \leq n} \mathsf{Reg}_i^{(T)}$, where $\mathsf{Reg}_i^{(T)}$ is the regret of the $i$th player.*

This connection holds for simultaneous updates. It is unclear if and how it can be extended under alternating updates. For the special case of potential games and RM$^+$, which is our main focus here, we are indeed able to establish convergence to the set of CCEs even under alternating updates by bounding the path length of the players' strategies (Remark C.9).

**Other notions of regret**  Section 2.2 introduced the usual notion of regret used in online linear optimization. For a constrained optimization problem with respect to a differentiable function $u$, we have $\mathsf{Reg}^{(T)} = \max_{\boldsymbol{x}' \in \mathcal{X}} \sum_{t=1}^T \langle \boldsymbol{x}' - \boldsymbol{x}^{(t)}, \nabla_{\boldsymbol{x}} u(\boldsymbol{x}^{(t)}) \rangle$. This is a linearized version of $\max_{\boldsymbol{x}' \in \mathcal{X}} \sum_{t=1}^T (u(\boldsymbol{x}') - u(\boldsymbol{x}^{(t)}))$. Minimizing the latter notion is computationally intractable unless one places restrictive assumptions on $u$. Hazan et al. (2017) (cf. Angelopoulos et al., 2025) introduced the notion of "local regret," and showed that having sublinear local regret implies that a randomly selected iterate will be an approximate stationary point; this is in stark contrast to regret as defined in Section 2.2 (Proposition 3.2).

**Discounting**  Next, we spell out regret matching$^+$ with discounting (DRM$^+$; Algorithm 3). The only difference from RM$^+$ is that the regret vector is multiplied by a discounting coefficient $\alpha^{(t)} \in (0, 1]$ in every round $t \in [T]$ (Algorithm 3); the special case where $\alpha^{(t)} = 1$ for all $t \in [T]$ is RM$^+$.

---

**Algorithm 3:** Regret matching$^+$ with discounting (DRM$^+$)

---
1 **Input**: discounting coefficients $(\alpha^{(1)}, \ldots, \alpha^{(T)}) \in (0, 1]^T$;
2 Initialize cumulative regrets $\boldsymbol{r}^{(0)} := \boldsymbol{0}$;
3 Initialize strategy $\boldsymbol{x}^{(1)} \in \Delta(\mathcal{A})$;
4 **for** $t = 1, \ldots, T$ **do**
5     Set $\boldsymbol{\theta}^{(t)} \leftarrow \boldsymbol{r}^{(t-1)}$;
6     **if** $\boldsymbol{\theta}^{(t)} \neq \boldsymbol{0}$ **then**
7        Compute $\boldsymbol{x}^{(t)} \leftarrow \boldsymbol{\theta}^{(t)} / \|\boldsymbol{\theta}^{(t)}\|_1$;
8     **else**
9        $\boldsymbol{x}^{(t)} \leftarrow \boldsymbol{x}^{(t-1)}$;
10     Output strategy $\boldsymbol{x}^{(t)} \in \Delta(\mathcal{A})$ ;
11     Observe utility $\boldsymbol{u}^{(t)} \in \mathbb{R}^{\mathcal{A}}$;
12     $\boldsymbol{r}^{(t)} \leftarrow \alpha^{(t)} [\boldsymbol{r}^{(t-1)} + \boldsymbol{u}^{(t)} - \langle \boldsymbol{x}^{(t)}, \boldsymbol{u}^{(t)} \rangle \boldsymbol{1}]^+$;

---

## C  OMITTED PROOFS

This section provides the proofs missing from the main body. We begin by stating a simple lemma that bounds the regret of RM$^+$, implying Proposition 2.3; we will then adapt it to account for discounting per Algorithm 3.

**Lemma C.1** (Regret vector upper bound). *For any time $t \in [T]$, RM$^+$ guarantees $\|\boldsymbol{r}^{(t)}\|_2^2 \leq \|\boldsymbol{r}^{(t-1)}\|_2^2 + \|\boldsymbol{g}^{(t)}\|_2^2$, where $\boldsymbol{g}^{(t)} := \boldsymbol{u}^{(t)} - \langle \boldsymbol{x}^{(t)}, \boldsymbol{u}^{(t)} \rangle$ is the instantaneous regret at time $t$.*

*Proof.* By definition of RM$^+$, $\langle \boldsymbol{r}^{(t-1)}, \boldsymbol{g}^{(t)} \rangle = \langle \boldsymbol{x}^{(t)}, \boldsymbol{g}^{(t)} \rangle = 0$ since $\boldsymbol{x}^{(t)} \propto \boldsymbol{r}^{(t-1)}$. Thus,

$$\|\boldsymbol{r}^{(t)}\|_2^2 = \|[\boldsymbol{r}^{(t-1)} + \boldsymbol{g}^{(t)}]^+\|_2^2 \leq \|\boldsymbol{r}^{(t-1)} + \boldsymbol{g}^{(t)}\|_2^2 = \|\boldsymbol{r}^{(t-1)}\|_2^2 + \|\boldsymbol{g}^{(t)}\|_2^2,$$

by orthogonality. $\qquad \square$

As a result, the telescopic summation yields $\|\boldsymbol{r}^{(T)}\|_2^2 \leq \sum_{t=1}^T \|\boldsymbol{g}^{(t)}\|_2^2 \leq mT$ since $\|\boldsymbol{g}^{(t)}\|_\infty \leq 1$ (by the assumption that the range of the utilities is bounded by 1). It is worth noting that a similar proof works for RM. We now adapt Lemma C.1 for DRM$^+$.

**Lemma C.2.** *For any time $t \in [T]$, DRM$^+$ guarantees $\|\boldsymbol{r}^{(t)}\|_2^2 \leq (\alpha^{(t)})^2 (\|\boldsymbol{r}^{(t-1)}\|_2^2 + \|\boldsymbol{g}^{(t)}\|_2^2)$.*

*Proof.* As before, $\langle r^{(t-1)}, g^{(t)} \rangle = \langle x^{(t)}, g^{(t)} \rangle = 0$ since $x^{(t)} \propto r^{(t-1)}$. Thus,

$$\|r^{(t)}\|_2^2 = (\alpha^{(t)})^2 \|[r^{(t-1)} + g^{(t)}]^+\|_2^2 \le (\alpha^{(t)})^2 \|r^{(t-1)} + g^{(t)}\|_2^2 \le (\alpha^{(t)})^2 (\|r^{(t-1)}\|_2^2 + \|g^{(t)}\|_2^2).$$

$\square$

A direct consequence is the following bound on the regret vector.

**Corollary C.3.** *For any time $t \in [T]$, DRM$^+$ guarantees*

$$\|r^{(t)}\|_2^2 \le (\alpha^{(t)})^2 \|g^{(t)}\|_2^2 + (\alpha^{(t)}\alpha^{(t-1)})^2 \|g^{(t-1)}\|_2^2 + \cdots + \left( \prod_{\tau=1}^{t} \alpha^{(\tau)} \right)^2 \|g^{(1)}\|_2^2. \quad (7)$$

*In particular, if $\alpha^{(t)} = 1 - \gamma$ for some constant $\gamma \in (0, 1)$, it follows that $\|r^{(T)}\|_2 \le \sqrt{m/\gamma}$.*

*Proof.* The first part of the claim follows by unfolding the bound of Lemma C.2. For the second part, using (7) and the fact that $\|g^{(t)}\|_2^2 \le m$ for any $t$, we have

$$\|r^{(t)}\|_2^2 \le m \left( (1-\gamma)^2 + (1-\gamma)^4 + \cdots + (1-\gamma)^{2t} \right) \le m \frac{(1-\gamma)^2}{1 - (1-\gamma)^2} \le m \frac{1}{\gamma}.$$

$\square$

## C.1 Proofs from Section 3

We continue with the proofs from Section 3. We first establish that RM$^+$ enjoys a one-step improvement property when the utility is linear.

**Lemma 3.3** (One-step improvement of RM$^+$). *For any $r \in \mathbb{R}_{\ge 0}^{\mathcal{A}}$ and $u \in \mathbb{R}^{\mathcal{A}}$, we define $x := r/\|r\|_1$; if $r = 0$, $x \in \Delta(\mathcal{A})$ can be arbitrary. If $r' := [r + u - \langle x, u \rangle 1]^+ \ne 0$ and $x' := r'/\|r'\|_1$, then*

$$\langle x' - x, u \rangle \ge \frac{1}{\|r'\|_1} \left( \max_{a \in \mathcal{A}} u[a] - \langle x, u \rangle \right)^2 = \frac{1}{\|r'\|_1} \mathrm{BRGap}(x, u)^2. \quad (4)$$

*If $r' = 0$, then $\langle x, u \rangle = \langle x', u \rangle \ge \max_{a \in \mathcal{A}} u[a]$.*

*Proof.* First, if $r' = 0$, it follows that $r + u - \langle x, u \rangle 1 \le 0$, where the inequality is to be taken coordinate-wise. Since $r \ge 0$, we have $\langle x, u \rangle \ge u[a]$ for all $a \in \mathcal{A}$, as claimed.

We now assume $r' \ne 0$. If $r = 0$, we have $r' = [u - \langle x, u \rangle 1]^+$. (4) can then be equivalently expressed as

$$\sum_{a \in \mathcal{A}} r'[a](u[a] - \langle x, u \rangle) \ge \left( \max_{a \in \mathcal{A}} u[a] - \langle x, u \rangle \right)^2,$$

which holds since $r' = [u - \langle x, u \rangle 1]^+$. So we can assume $r \ne 0$. We define $\delta := r' - r$. (4) can be expressed as

$$\frac{\sum_{a \in \mathcal{A}} (r[a] + \delta[a]) u[a]}{\sum_{a' \in \mathcal{A}} (r[a'] + \delta[a'])} \ge \frac{\sum_{a \in \mathcal{A}} r[a] u[a]}{\sum_{a' \in \mathcal{A}} r[a']} + \frac{(\max_{a \in \mathcal{A}} u[a] - \langle x, u \rangle)^2}{\sum_{a' \in \mathcal{A}} (r[a'] + \delta[a'])}.$$

Equivalently,

$$\sum_{a' \in \mathcal{A}} r[a'] \sum_{a \in \mathcal{A}} (r[a] + \delta[a]) u[a] \ge \sum_{a \in \mathcal{A}} r[a] \sum_{a' \in \mathcal{A}} (r[a'] + \delta[a']) u[a]$$

$$+ \sum_{a' \in \mathcal{A}} r[a'] \left( \max_{a \in \mathcal{A}} u[a] - \langle x, u \rangle \right)^2.$$

This in turn is equivalent to

$$\sum_{a' \in \mathcal{A}} r[a'] \sum_{a \in \mathcal{A}} \delta[a] u[a] \ge \sum_{a \in \mathcal{A}} r[a] \sum_{a' \in \mathcal{A}} \delta[a'] u[a] + \sum_{a' \in \mathcal{A}} r[a'] \left( \max_{a \in \mathcal{A}} u[a] - \langle x, u \rangle \right)^2$$

$$= \sum_{a' \in \mathcal{A}} \delta[a'] \sum_{a \in \mathcal{A}} r[a] \langle x, u \rangle + \sum_{a' \in \mathcal{A}} r[a'] \left( \max_{a \in \mathcal{A}} u[a] - \langle x, u \rangle \right)^2.$$

Rearranging,

$$\sum_{a' \in \mathcal{A}} \boldsymbol{r}[a'] \sum_{a \in \mathcal{A}} \boldsymbol{\delta}[a](\boldsymbol{u}[a] - \langle \boldsymbol{x}, \boldsymbol{u} \rangle) \geq \sum_{a' \in \mathcal{A}} \boldsymbol{r}[a'] \left( \max_{a \in \mathcal{A}} \boldsymbol{u}[a] - \langle \boldsymbol{x}, \boldsymbol{u} \rangle \right)^2.$$

Now, for any $a \in \mathcal{A}$ such that $\boldsymbol{u}[a] - \langle \boldsymbol{x}, \boldsymbol{u} \rangle \geq 0$, it follows that $\boldsymbol{\delta}[a] = \boldsymbol{u}[a] - \langle \boldsymbol{x}, \boldsymbol{u} \rangle \geq 0$; on the other hand, for $a \in \mathcal{A}$ such that $\boldsymbol{u}[a] - \langle \boldsymbol{x}, \boldsymbol{u} \rangle < 0$, we have $\boldsymbol{\delta}[a] \leq 0$. That is, $\boldsymbol{\delta}[a](\boldsymbol{u}[a] - \langle \boldsymbol{x}, \boldsymbol{u} \rangle) \geq 0$, and the claim follows. $\square$

We will now show that Lemma 3.3 is, in a certain sense, tight. We consider a simple linear maximization over the simplex. If the regret vector of $\text{RM}^+$ can be initialized arbitrarily, as is the premise in Lemma 3.3, we make the following observation.

**Lemma C.4.** *Consider a utility vector $\boldsymbol{u} \in \mathbb{R}^{\mathcal{A}}$ and some initial regret vector $\mathbb{R}_{\geq 0}^{\mathcal{A}} \ni \boldsymbol{r}^{(1)} \neq \boldsymbol{0}$. If $\boldsymbol{x}^{(1)} = \boldsymbol{r}^{(1)}/\|\boldsymbol{r}^{(1)}\|_1$ and $\epsilon = \max_{a \in \mathcal{A}} \boldsymbol{u}[a] - \langle \boldsymbol{x}^{(1)}, \boldsymbol{u} \rangle$ is the initial best-response gap, it takes at least $\|\boldsymbol{r}^{(1)}\|_1/2\epsilon$ iterations for $\text{RM}^+$ to reach a point $\boldsymbol{x}^{(t)}$ with best-response gap at most $\epsilon/2$.*

Indeed, we consider the two-dimensional problem in which $\boldsymbol{u} = (1 - \epsilon, 1)$ and $\boldsymbol{r}^{(1)} = (\|\boldsymbol{r}^{(1)}\|_1, 0)$. To incur a best-response gap of at most $\epsilon/2$, the player needs to allot a probability mass of at least $1/2$ to the second action. In the meantime, the decrement of the first coordinate of $\boldsymbol{r}^{(t)}$ will be at most $\epsilon$ while the increment of the second coordinate of $\boldsymbol{r}^{(t)}$ will be at most $\epsilon$. But, for the algorithm to terminate, it must be the case that the second coordinate of $\boldsymbol{r}^{(t)}$ is at least as large as the first coordinate of $\boldsymbol{r}^{(t)}$, leading to Lemma C.4.

Now, given that $\langle \boldsymbol{x}^{(t)} - \boldsymbol{x}^{(1)}, \boldsymbol{u} \rangle \leq \epsilon$, Lemma C.4 matches the bound obtained for this problem through Lemma 3.3 in the regime where $\|\boldsymbol{r}^{(1)}\|_1$ is at least as large as $1/\epsilon$ (so that the norm of $\boldsymbol{r}^{(t)}$ is within a constant factor of that of $\boldsymbol{r}^{(1)}$, by Lemma C.1).

Taking this argument a step further, if we have a regret bound of the form $\|\boldsymbol{r}^{(t)}\|_1 \leq \|\boldsymbol{r}\|_1 = O_t(1)$, which holds when the utility is fixed, Lemma 3.3 implies that $2\|\boldsymbol{r}\|_1 + 4\|\boldsymbol{r}\|_1 + \cdots + \|\boldsymbol{r}\|_1/\epsilon = O_\epsilon(1/\epsilon)$ iterations suffice for $\text{RM}^+$ to have a best-response gap at most $\epsilon$ when facing a fixed utility; this follows by applying Lemma 3.3 first for all iterations in which the best-response gap is at least $1/2$, then for all iterations in which it is at least $1/4$, and so forth.

**Analysis of non-lazy alternating $\text{RM}^+$** In the main body, we used Lemma 3.3 to argue that *lazy* alternating $\text{RM}^+$ converges in potential games (Theorem 3.4). One caveat of the lazy version of alternation is that it requires knowing the desired precision $\epsilon$ ahead of time. We will now extend the analysis to encompass the usual version of alternation, albeit at the cost of introducing a further dependence on the iteration bound.

To begin with, we first state a direct refinement of Lemma 3.3 that we rely on; this refinement will also be useful in the more general setting of constrained optimization.

**Lemma C.5** (Refinement of Lemma 3.3). *Under the preconditions of Lemma 3.3,*

$$\langle \boldsymbol{x}' - \boldsymbol{x}, \boldsymbol{u} \rangle \geq \frac{1}{\|\boldsymbol{r}'\|_1} \|\boldsymbol{r} - \boldsymbol{r}'\|_2^2. \tag{8}$$

In particular, (8) implies (4) since $\|\boldsymbol{r} - \boldsymbol{r}'\|_2^2 \geq \|\boldsymbol{r} - \boldsymbol{r}'\|_\infty^2 \geq (\max_{a \in \mathcal{A}} \boldsymbol{u}[a] - \langle \boldsymbol{x}, \boldsymbol{u} \rangle)^2$, by definition of $\boldsymbol{r}'$. The proof of Lemma C.5 is identical to that of Lemma 3.3.

The next elementary lemma shows that, so long as the norm of the regret vector is not too small, closeness in regrets implies closeness in strategies.

**Lemma C.6.** *For $\mathbb{R}_{\geq 0}^{\mathcal{A}} \ni \boldsymbol{r}, \boldsymbol{r}' \neq \boldsymbol{0}$, let $\boldsymbol{x} := \boldsymbol{r}/\|\boldsymbol{r}\|_1$ and $\boldsymbol{x}' := \boldsymbol{r}'/\|\boldsymbol{r}'\|_1$. Then*

$$\|\boldsymbol{x} - \boldsymbol{x}'\|_1 \leq \|\boldsymbol{r} - \boldsymbol{r}'\|_1 \left( \frac{1}{\|\boldsymbol{r}\|_1} + \frac{1}{\|\boldsymbol{r}'\|_1} \right).$$

*Proof.* The term $\boldsymbol{x}[a] - \boldsymbol{x}'[a]$ can be expressed, for any $a \in \mathcal{A}$, as

$$\frac{\boldsymbol{r}[a]}{\sum_{a' \in \mathcal{A}} \boldsymbol{r}[a']} - \frac{\boldsymbol{r}'[a]}{\sum_{a' \in \mathcal{A}} \boldsymbol{r}'[a']} = \frac{\sum_{a' \in \mathcal{A}} (\boldsymbol{r}[a]\boldsymbol{r}'[a'] - \boldsymbol{r}'[a]\boldsymbol{r}[a'])}{\|\boldsymbol{r}\|_1 \|\boldsymbol{r}'\|_1}$$

$$= \frac{\sum_{a' \in \mathcal{A}} (\boldsymbol{r}[a](\boldsymbol{r}'[a'] - \boldsymbol{r}[a']) + \boldsymbol{r}[a'](\boldsymbol{r}[a] - \boldsymbol{r}'[a]))}{\|\boldsymbol{r}\|_1 \|\boldsymbol{r}'\|_1},$$

and the claim follows. $\qquad\square$

With those two helper lemmas in hand, we are now ready to analyze (non-lazy) alternating RM$^+$ in potential games.

**Theorem C.7.** *In any potential game with utilities in $[0, 1]$, alternating RM$^+$ requires at most $2\lceil \frac{625m^4 n^4 \Phi_{\text{range}}^2}{\delta^4 \epsilon^4} \rceil$ rounds to converge to an $\epsilon$-Nash equilibrium, where $\delta_i \coloneqq \mathsf{BRGap}_i(\boldsymbol{x}_i^{(1)}, \boldsymbol{u}_i^{(1)}) > 0$ and $\delta \coloneqq \min_{1 \leq i \leq n} \delta_i$.*

The assumption that $\delta_i > 0$ is without any loss in the following sense. The analysis requires that $\|\boldsymbol{r}_i^{(t)}\|_2 > 0$ for all $i \in [n]$ and any sufficiently large $t$. By Lemma 3.8, it suffices if a player incurs a positive best-response gap *at some* time. In the contrary case, if a player always has zero best-response gap, it will always play the same strategy (by definition of RM$^+$ in Algorithm 2), thereby reducing to a potential game with $n - 1$ players. We further remark that, in accordance with Theorem 3.4, the bound in Theorem C.7 can be similarly parameterized in terms of the maximum regret incurred by a player. A more pedantic point about Theorem C.7 is that utilities are taken to be in $[0, 1]$, while in the rest of the paper we only assume that the range is bounded by 1; this innocuous assumption is used in Claim C.8 below.

*Proof of Theorem C.7.* By Lemma C.5, the telescopic summation after $T$ rounds yields

$$\Phi_{\text{range}} \geq \sum_{t=1}^{T} \sum_{i=1}^{n} \frac{1}{\|\boldsymbol{r}_i^{(t)}\|_1} \|\boldsymbol{r}_i^{(t)} - \boldsymbol{r}_i^{(t-1)}\|_2^2 \geq \sum_{t=1}^{T} \frac{1}{\max_i \|\boldsymbol{r}_i^{(t)}\|_1} \sum_{i=1}^{n} \|\boldsymbol{r}_i^{(t)} - \boldsymbol{r}_i^{(t-1)}\|_2^2. \quad (9)$$

Since $\|\boldsymbol{r}_i^{(t)}\|_1 \leq m\sqrt{t}$ for any player $i \in [n]$, it follows that $\sum_{t=1}^{T} \sum_{i=1}^{n} \|\boldsymbol{r}_i^{(t+1)} - \boldsymbol{r}_i^{(t)}\|_2^2 \leq m\Phi_{\text{range}}\sqrt{T}$. As a result, after at most $2\lceil m^2 \Phi_{\text{range}}^2 / \epsilon^4 \rceil$ rounds, there will be a time $t$ such that $\sum_{i=1}^{n} \|\boldsymbol{r}_i^{(t)} - \boldsymbol{r}_i^{(t-1)}\|_2^2 + \sum_{i=1}^{n} \|\boldsymbol{r}_i^{(t+1)} - \boldsymbol{r}_i^{(t)}\|_2^2 \leq \epsilon^2$, which in turn implies $\|\boldsymbol{r}_i^{(t)} - \boldsymbol{r}_i^{(t-1)}\|_2, \|\boldsymbol{r}_i^{(t+1)} - \boldsymbol{r}_i^{(t)}\|_2 \leq \epsilon$ for all $i \in [n]$. Now, by assumption, we know that $\|\boldsymbol{r}_i^{(1)}\|_2 \geq \delta_i > 0$ for all $i \in [n]$. By the monotonicity property of the regret vector (Lemma 3.8, proven in the sequel), it follows that $\|\boldsymbol{r}_i^{(t)}\|_2 \geq \delta_i > 0$ for all $t \in [T+1]$ and $i \in [n]$. Consequently, applying Lemma C.6, we have $\|\boldsymbol{x}_i^{(t+1)} - \boldsymbol{x}_i^{(t)}\|_1 \leq 2\|\boldsymbol{r}_i^{(t+1)} - \boldsymbol{r}_i^{(t)}\|_1 / \delta_i \leq 2\sqrt{m}\epsilon / \delta_i$ since $\|\cdot\|_1 \leq \sqrt{m}\|\cdot\|_2$. Next, we will make use of the following simple claim.

**Claim C.8.** *Consider a normal-form game with utilities in $[0, 1]$. For any two joint strategies $(\boldsymbol{x}_1, \ldots, \boldsymbol{x}_n)$ and $(\boldsymbol{x}_1', \ldots, \boldsymbol{x}_n')$, it holds that $\|\boldsymbol{u}_i(\boldsymbol{x}_{-i}) - \boldsymbol{u}_i(\boldsymbol{x}_{-i}')\|_\infty \leq \sum_{i' \neq i} \|\boldsymbol{x}_{i'} - \boldsymbol{x}_{i'}'\|_1$ for any player $i \in [n]$.*

*Proof.* We fix a player $i \in [n]$. We have

$$\|\boldsymbol{u}_i(\boldsymbol{x}_{-i}) - \boldsymbol{u}_i(\boldsymbol{x}_{-i}')\|_\infty = \left| \sum_{\boldsymbol{a} \in \mathcal{A}_1 \times \cdots \times \mathcal{A}_n} u_i(\cdot, \boldsymbol{a}_{-i}) \left( \prod_{i' \neq i} \boldsymbol{x}_{i'}[a_{i'}] - \prod_{i' \neq i} \boldsymbol{x}_{i'}'[a_{i'}] \right) \right|$$

$$\leq \left| \sum_{\boldsymbol{a} \in \mathcal{A}_1 \times \cdots \times \mathcal{A}_n} \left( \prod_{i' \neq i} \boldsymbol{x}_{i'}[a_{i'}] - \prod_{i' \neq i} \boldsymbol{x}_{i'}'[a_{i'}] \right) \right| \quad (10)$$

$$\leq \sum_{i' \neq i} \|\boldsymbol{x}_{i'} - \boldsymbol{x}_{i'}'\|_1, \quad (11)$$

where (10) uses triangle inequality together with the assumption that $|u_i(\cdot)| \leq 1$, and (11) uses a bound on the total variation distance of a product distribution in terms of the sum of the total variation distances of its marginals (Hoeffding & Wolfowitz, 1958). $\qquad\square$

Using this lemma, we now observe that $\|\boldsymbol{u}_i^{(t)} - \boldsymbol{u}_i(\boldsymbol{x}_{-i}^{(t)})\|_\infty \leq \sum_{i' < i} \|\boldsymbol{x}_{i'}^{(t+1)} - \boldsymbol{x}_{i'}^{(t)}\|_1 \leq \sum_{i' \neq i} \|\boldsymbol{x}_{i'}^{(t+1)} - \boldsymbol{x}_{i'}^{(t)}\|_1 \leq 2n\sqrt{m}\epsilon/\delta$ for each $i \in [n]$, where $\delta = \min_{1 \leq i \leq n} \delta_i$. Furthermore, in view of the fact that $\|\boldsymbol{r}_i^{(t)} - \boldsymbol{r}_i^{(t-1)}\|_2 \leq \epsilon$, it follows that $\|\boldsymbol{u}_i^{(t)}\|_\infty - \langle \boldsymbol{x}_i^{(t)}, \boldsymbol{u}_i^{(t)} \rangle \leq \epsilon$. Thus,

$$
\begin{aligned}
\|\boldsymbol{u}_i(\boldsymbol{x}_{-i}^{(t)})\|_\infty - \langle \boldsymbol{x}_i^{(t)}, \boldsymbol{u}_i(\boldsymbol{x}_{-i}^{(t)}) \rangle &= \|\boldsymbol{u}_i^{(t)}\|_\infty - \langle \boldsymbol{x}_i^{(t)}, \boldsymbol{u}_i^{(t)} \rangle \\
&\quad + \|\boldsymbol{u}_i(\boldsymbol{x}_{-i}^{(t)})\|_\infty - \|\boldsymbol{u}_i^{(t)}\|_\infty + \langle \boldsymbol{x}_i^{(t)}, \boldsymbol{u}_i^{(t)} - \boldsymbol{u}_i(\boldsymbol{x}_{-i}^{(t)}) \rangle \\
&\leq \|\boldsymbol{u}_i^{(t)}\|_\infty - \langle \boldsymbol{x}_i^{(t)}, \boldsymbol{u}_i^{(t)} \rangle + 2\|\boldsymbol{u}_i^{(t)} - \boldsymbol{u}_i(\boldsymbol{x}_{-i}^{(t)})\|_\infty \qquad (12) \\
&\leq \epsilon \left( 1 + \frac{4n\sqrt{m}}{\delta} \right) \leq \epsilon \left( 5\frac{n\sqrt{m}}{\delta} \right).
\end{aligned}
$$

where (12) follows from the fact that $\langle \boldsymbol{x}_i^{(t)}, \boldsymbol{u}_i^{(t)} - \boldsymbol{u}_i(\boldsymbol{x}_{-i}^{(t)}) \rangle \leq \|\boldsymbol{x}_i^{(t)}\|_1 \|\boldsymbol{u}_i^{(t)} - \boldsymbol{u}_i(\boldsymbol{x}_{-i}^{(t)})\|_\infty \leq \|\boldsymbol{u}_i^{(t)} - \boldsymbol{u}_i(\boldsymbol{x}_{-i}^{(t)})\|_\infty$ since $\|\boldsymbol{x}_i^{(t)}\|_1 = 1$. We conclude that $(\boldsymbol{x}_1^{(t)}, \ldots, \boldsymbol{x}_n^{(t)})$ is an $\epsilon^{(5n\sqrt{m}/\delta)}$-Nash equilibrium; rescaling $\epsilon$ leads to the claim. $\qquad \square$

*Remark* C.9 (Convergence to CCE under alternating updates). The folk connection linking no-regret learning and coarse correlated equilibria (Proposition B.2) is predicated on the dynamics being executed in a simultaneous fashion. We observe that, in potential games, even alternating RM$^+$ converges to the set of CCEs in the following sense. Lemma C.6 together with (9) imply that $\sum_{t=1}^T \sum_{i=1}^n \|\boldsymbol{x}_i^{(t+1)} - \boldsymbol{x}_i^{(t)}\|_1^2 = O_T(\sqrt{T})$. The Cauchy-Schwarz inequality in turn yields

$$
\sum_{t=1}^T \sum_{i=1}^n \|\boldsymbol{x}_i^{(t+1)} - \boldsymbol{x}_i^{(t)}\|_1 \leq \sqrt{\sum_{t=1}^T \left( \sum_{i=1}^n \|\boldsymbol{x}_i^{(t+1)} - \boldsymbol{x}_i^{(t)}\|_1 \right)^2 \sum_{t=1}^T 1^2} = O_T(T^{3/4}). \qquad (13)
$$

Now, by the fact that RM$^+$ has the no-regret property (Proposition 2.3), $\max_{\boldsymbol{x}_i' \in \Delta(\mathcal{A}_i)} \sum_{t=1}^T \langle \boldsymbol{x}_i' - \boldsymbol{x}_i^{(t)}, \boldsymbol{u}_i^{(t)} \rangle = O_T(\sqrt{T})$. Further, by Claim C.8 and (13), $\sum_{t=1}^T \|\boldsymbol{u}_i(\boldsymbol{x}_{-i}^{(t)}) - \boldsymbol{u}_i^{(t)}\|_\infty \leq \sum_{t=1}^T \sum_{i' \neq i} \|\boldsymbol{x}_{i'}^{(t+1)} - \boldsymbol{x}_{i'}^{(t)}\|_1 = O_T(T^{3/4})$. As a result, we conclude that, for any player $i \in [n]$,

$$
\max_{\boldsymbol{x}_i' \in \Delta(\mathcal{A}_i)} \sum_{t=1}^T \langle \boldsymbol{x}_i' - \boldsymbol{x}_i^{(t)}, \boldsymbol{u}_i(\boldsymbol{x}_{-i}^{(t)}) \rangle = O_T(T^{3/4}).
$$

This implies that the correlated distribution $\frac{1}{T} \sum_{t=1}^T \bigotimes_{i=1}^n \boldsymbol{x}_i^{(t)}$ is an $\epsilon$-CCE for some $\epsilon = O_T(T^{-1/4})$ (Proposition B.2); here, $\bigotimes_{i=1}^n \boldsymbol{x}_i^{(t)}$ denotes the product distribution induced by $(\boldsymbol{x}_1^{(t)}, \ldots, \boldsymbol{x}_n^{(t)})$.

Moving on, a further implication of Lemma C.4 is that having a large regret vector can slow down RM$^+$. Employing discounting, in the form of DRM$^+$ (Algorithm 3), addresses this deficiency in potential games, as we prove below. It is worth stressing that while discounting is sensible when employing alternating RM$^+$ in potential games, this is not the case in more general constrained optimization problems; there, having a regret vector with a *small* norm can impede convergence (*cf.* Lemma 3.7).

**Corollary 3.5.** *In any potential game, $\epsilon$-lazy alternating DRM$^+$ with discount factor $1 - \gamma \in (0, 1)$ requires at most $1 + \frac{m\Phi_{\mathrm{range}}}{\epsilon^2 \sqrt{\gamma}}$ rounds to converge to an $\epsilon$-Nash equilibrium.*

*Proof.* Lemma 3.3 can be directly adjusted for DRM$^+$: the updated strategy $\boldsymbol{x}' = \boldsymbol{r}'/\|\boldsymbol{r}'\|_1$, where $\boldsymbol{r}' = \alpha[\boldsymbol{r} + \boldsymbol{u} - \langle \boldsymbol{x}, \boldsymbol{u} \rangle \mathbf{1}]^+ \neq \mathbf{0}$, remains the same since we only rescaled the regret vector by $\alpha$. That is, for any $\boldsymbol{u} \in \mathbb{R}^\mathcal{A}$,

$$
\langle \boldsymbol{x}' - \boldsymbol{x}, \boldsymbol{u} \rangle \geq \frac{1}{\|\boldsymbol{r}'\|_1} \left( \max_{a \in \mathcal{A}} \boldsymbol{u}[a] - \langle \boldsymbol{x}, \boldsymbol{u} \rangle \right)^2.
$$

The key difference compared to Theorem 3.4 is that we now maintain the invariance $\|\boldsymbol{r}_i^{(t)}\|_2 \leq \sqrt{m/\gamma}$ for all $i \in [n]$ and $t \in [T]$ (Corollary C.3). Consequently,

$$
\Phi_{\mathrm{range}} \geq \sum_{t=1}^T \sum_{i=1}^n \frac{1}{\|\boldsymbol{r}_i^{(t)}\|_1} \mathrm{BRGap}_i(\boldsymbol{x}_i^{(t)}, \boldsymbol{u}_i^{(t)})^2 \mathbb{1}\{\mathrm{BRGap}_i(\boldsymbol{x}_i^{(t)}, \boldsymbol{u}_i^{(t)}) > \epsilon\}.
$$

If in every round $t \in [T]$ there is a player $i \in [n]$ such that $\mathsf{BRGap}_i(\boldsymbol{x}_i^{(t)}, \boldsymbol{u}_i^{(t)}) > \epsilon$, we have $\Phi_{\mathsf{range}} \geq \sum_{t=1}^{T} (\sqrt{\gamma}/m)\epsilon^2$, so $T \leq \frac{m\Phi_{\mathsf{range}}}{\epsilon^2 \sqrt{\gamma}}$. This concludes the proof. $\qquad\square$

Unlike $\mathsf{RM}^+$ and $\mathsf{DRM}^+$, $\mathsf{RM}$ only has a *conditional* one-step improvement because the regret vector can have negative coordinates (*cf.* Grand-Clément & Kroer, 2024).

**Lemma 3.6.** *For any $\boldsymbol{r} \in \mathbb{R}_{\geq 0}^{\mathcal{A}}$ and $\boldsymbol{u} \in \mathbb{R}^{\mathcal{A}}$, we define $\boldsymbol{x} := \theta/\|\theta\|_1$, where $\theta := \max(\boldsymbol{r}, \boldsymbol{0})$; if $\theta = \boldsymbol{0}$, $\boldsymbol{x} \in \Delta(\mathcal{A})$ can be arbitrary. If $\boldsymbol{r}' := \boldsymbol{r} + \boldsymbol{u} - \langle \boldsymbol{x}, \boldsymbol{u} \rangle \boldsymbol{1}$ and $\boldsymbol{x}' := \theta'/\|\theta'\|_1$, where $\theta' = \max(\boldsymbol{r}', \boldsymbol{0}) \neq \boldsymbol{0}$, then $\langle \boldsymbol{x}' - \boldsymbol{x}, \boldsymbol{u} \rangle \geq \frac{1}{\|\theta'\|_1} \|\theta' - \theta\|_2^2 \geq \frac{1}{\|\theta'\|_1} (\max_{a \in \mathcal{A}} \boldsymbol{u}[a] - \langle \boldsymbol{x}, \boldsymbol{u} \rangle)^2 \mathbb{1}\{\boldsymbol{r}[a] \geq 0\}$, where $a \in \arg\max_{a' \in \mathcal{A}} \boldsymbol{u}[a']$. If $\theta' = \boldsymbol{0}$, then $\langle \boldsymbol{x}, \boldsymbol{u} \rangle = \langle \boldsymbol{x}', \boldsymbol{u} \rangle \geq \boldsymbol{u}[a]$.*

*Proof.* We define $\delta := \theta' - \theta$. Following the proof of Lemma 3.3, it suffices to show that

$$\sum_{a' \in \mathcal{A}} \theta[a'] \sum_{a \in \mathcal{A}} \delta[a](\boldsymbol{u}[a] - \langle \boldsymbol{x}, \boldsymbol{u} \rangle) \geq \sum_{a' \in \mathcal{A}} \theta[a'] \left( \max_{a \in \mathcal{A}} \boldsymbol{u}[a] - \langle \boldsymbol{x}, \boldsymbol{u} \rangle \right)^2 \mathbb{1}\{\boldsymbol{r}[a] \geq 0\}. \quad (14)$$

For an action $a \in \mathcal{A}$, we consider the following cases.

- If $\boldsymbol{u}[a] - \langle \boldsymbol{x}, \boldsymbol{u} \rangle \geq 0$,

    - if $\boldsymbol{r}[a] \geq 0$, we have $\delta[a] = \boldsymbol{u}[a] - \langle \boldsymbol{x}, \boldsymbol{u} \rangle$.
    - If $\boldsymbol{r}[a] < 0$, it follows that $\delta[a] \geq 0$; in particular, $\delta[a] = 0$ if $\boldsymbol{r}'[a] \leq 0$ and $\delta[a] > 0$ otherwise. As a result, $\delta[a](\boldsymbol{u}[a] - \langle \boldsymbol{x}, \boldsymbol{u} \rangle) \geq 0$.

- If $\boldsymbol{u}[a] - \langle \boldsymbol{x}, \boldsymbol{u} \rangle < 0$,

    - if $\boldsymbol{r}[a] \leq 0$, we have $\delta[a] = 0$ since $\theta[a] = 0 = \theta'[a]$.
    - if $\boldsymbol{r}[a] > 0$, it follows that $\delta[a] < 0$. Again, we have $\delta[a](\boldsymbol{u}[a] - \langle \boldsymbol{x}, \boldsymbol{u} \rangle) \geq 0$.

Combining those items, (14) follows. $\qquad\square$

We now turn to the more general setting of constrained optimization. First, combining Lemmas C.5 and C.6 that were proven earlier, we formally show that $\mathsf{RM}^+$ improves the value of the underlying function provided that the norm of the regret vector is not too small.

**Lemma 3.7.** *Let $u$ be an $L$-smooth function over $\Delta(\mathcal{A})$. For any $\boldsymbol{r} \in \mathbb{R}_{\geq 0}^{\mathcal{A}}$ with $\boldsymbol{r} \neq \boldsymbol{0}$, we define $\boldsymbol{x} := \boldsymbol{r}/\|\boldsymbol{r}\|_1$. Further, let $\boldsymbol{r}' := [\boldsymbol{r} + \nabla u(\boldsymbol{x}) - \langle \boldsymbol{x}, \nabla u(\boldsymbol{x}) \rangle \boldsymbol{1}]^+ \neq \boldsymbol{0}$ and $\boldsymbol{x}' := \boldsymbol{r}'/\|\boldsymbol{r}'\|_1$. If $\|\boldsymbol{r}'\|_2 \geq \max\{2m, 9mL\}$, then $u(\boldsymbol{x}') - u(\boldsymbol{x}) \geq \frac{1}{2\|\boldsymbol{r}'\|_1} \left( \max_{\boldsymbol{x}^\star \in \Delta(\mathcal{A})} \langle \boldsymbol{x}^\star - \boldsymbol{x}, \nabla u(\boldsymbol{x}) \rangle \right)^2$.*

*Proof.* Using the quadratic bound for $u$, we have

$$u(\boldsymbol{x}') - u(\boldsymbol{x}) \geq \langle \nabla u(\boldsymbol{x}), \boldsymbol{x}' - \boldsymbol{x} \rangle - \frac{L}{2} \|\boldsymbol{x} - \boldsymbol{x}'\|_2^2$$

$$\geq \frac{1}{\|\boldsymbol{r}'\|_1} \|\boldsymbol{r} - \boldsymbol{r}'\|_2^2 - \frac{L}{2} \|\boldsymbol{r} - \boldsymbol{r}'\|_1^2 \left( \frac{1}{\|\boldsymbol{r}\|_1} + \frac{1}{\|\boldsymbol{r}'\|_1} \right)^2 \quad (15)$$

$$\geq \frac{1}{\|\boldsymbol{r}'\|_1} \|\boldsymbol{r} - \boldsymbol{r}'\|_2^2 - \frac{9mL}{2\|\boldsymbol{r}'\|_1^2} \|\boldsymbol{r} - \boldsymbol{r}'\|_2^2 \quad (16)$$

$$\geq \frac{1}{2\|\boldsymbol{r}'\|_1} \|\boldsymbol{r} - \boldsymbol{r}'\|_2^2, \quad (17)$$

where (15) uses the one-step improvement property (Lemma C.5) applied for $\boldsymbol{u} := \nabla u(\boldsymbol{x})$ together with Lemma C.6; (16) follows from the fact that $\|\boldsymbol{r}\|_1 \geq \|\boldsymbol{r}'\|_1 - m \geq \frac{1}{2}\|\boldsymbol{r}'\|_1$ since $\|\boldsymbol{r}'\|_1 \geq \|\boldsymbol{r}'\|_2 \geq 2m$ and $|\langle \boldsymbol{x} - \boldsymbol{x}', \nabla u(\boldsymbol{x}) \rangle| \leq 1$ for all $\boldsymbol{x}' \in \Delta(\mathcal{A})$ (per our normalization assumption); and (17) follows from the assumption that $\|\boldsymbol{r}'\|_1 \geq 9mL$. $\qquad\square$

To make use of Lemma 3.7, we next establish that the $\ell_2$ norm of the regret vector under $\mathsf{RM}^+$ is nondecreasing.

**Lemma 3.8.** *For any $t$, RM$^+$ guarantees $\|\boldsymbol{r}^{(t)}\|_2^2 \geq \|\boldsymbol{r}^{(t-1)}\|_2^2 + \|[\boldsymbol{g}^{(t)}]^+\|_2^2$, where $\boldsymbol{g}^{(t)} := \nabla u(\boldsymbol{x}^{(t)}) - \langle \nabla u(\boldsymbol{x}^{(t)}), \boldsymbol{x}^{(t)} \rangle \mathbf{1}$ is the instantaneous regret vector at round $t$.*

*Proof.* We have $\boldsymbol{r}^{(t)} - \boldsymbol{r}^{(t-1)} = \max(\boldsymbol{g}^{(t)}, -\boldsymbol{r}^{(t-1)})$ (element-wise), so

$$\|\boldsymbol{r}^{(t)} - \boldsymbol{r}^{(t-1)}\|_2 = \|\max(\boldsymbol{g}^{(t)}, -\boldsymbol{r}^{(t-1)})\|_2 \geq \|[\boldsymbol{g}^{(t)}]^+\|_2.$$

Further, $\langle \boldsymbol{r}^{(t-1)}, \boldsymbol{r}^{(t)} - \boldsymbol{r}^{(t-1)} \rangle = \langle \boldsymbol{r}^{(t-1)}, \max(\boldsymbol{g}^{(t)}, -\boldsymbol{r}^{(t-1)}) \rangle \geq \langle \boldsymbol{r}^{(t-1)}, \boldsymbol{g}^{(t)} \rangle = 0$, where we used the fact that $\boldsymbol{r}^t \geq \mathbf{0}$, coordinate-wise. Therefore,

$$\|\boldsymbol{r}^{(t)}\|_2^2 = \|\boldsymbol{r}^{(t)} - \boldsymbol{r}^{(t-1)} + \boldsymbol{r}^{(t-1)}\|_2^2 = \|\boldsymbol{r}^{(t)} - \boldsymbol{r}^{(t-1)}\|_2^2 + \|\boldsymbol{r}^{(t-1)}\|_2^2 + 2\langle \boldsymbol{r}^{(t-1)}, \boldsymbol{r}^{(t)} - \boldsymbol{r}^{(t-1)} \rangle$$
$$\geq \|\boldsymbol{r}^{(t-1)}\|_2^2 + \|[\boldsymbol{g}^{(t)}]^+\|_2^2,$$

as claimed. $\qquad\square$

Armed with Lemmas 3.7 and 3.8, we can now prove Theorem 3.9.

**Theorem 3.9.** *Let $u$ be an $L$-smooth function in $\Delta(\mathcal{A}) \subset \mathbb{R}^m$ with range $u_{range}$ and $R := \max\{2m, 9mL\}$. RM$^+$ requires at most $1 + \frac{(m(2u_{range}+R^2))^2}{\epsilon^4}$ rounds to reach an $\epsilon$-KKT point.*

*Proof.* Let $t_c \in [T]$ be the largest $t$ such that $\|\boldsymbol{r}^{(t)}\|_2 < \max\{2m, 9mL\} = R$. By Lemma 3.8,

$$\|\boldsymbol{r}^{(t)}\|_2^2 \geq \|\boldsymbol{r}^{(t-1)}\|_2^2 + \|[\boldsymbol{g}^{(t)}]_+\|_2^2 \geq \|\boldsymbol{r}^{(t-1)}\|_2^2 + \mathsf{KKTGap}(\boldsymbol{x}^{(t)})^2;$$

so,

$$\sum_{t=1}^{t_c} \mathsf{KKTGap}(\boldsymbol{x}^{(t)})^2 \leq \sum_{t=1}^{t_c} (\|\boldsymbol{r}^{(t)}\|_2^2 - \|\boldsymbol{r}^{(t-1)}\|_2^2) = \|\boldsymbol{r}^{(t_c)}\|_2^2 \leq R^2. \tag{18}$$

Further, for any $t \geq t_c + 1$, we have $\|\boldsymbol{r}^{(t)}\|_2 \geq R$ since the $\ell_2$ norm of the regret vector is nondecreasing (Lemma 3.8) and $\|\boldsymbol{r}^{(t_c+1)}\|_2 \geq R$. Thus, by Lemma 3.7,

$$\sum_{t=t_c+1}^{T} \frac{1}{2\|\boldsymbol{r}^{(t)}\|_1} \mathsf{KKTGap}(\boldsymbol{x}^{(t)})^2 \leq u(\boldsymbol{x}^{(T+1)}) - u(\boldsymbol{x}^{(t_c+1)}) \leq u_{range}. \tag{19}$$

Combining (18) and (19), together with the fact that $\|\boldsymbol{r}^{(t)}\|_1 \leq m\sqrt{t}$,

$$\sum_{t=1}^{T} \frac{1}{m\sqrt{t}} \mathsf{KKTGap}(\boldsymbol{x}^{(t)})^2 \leq 2u_{range} + R^2.$$

This leads to the claim. $\qquad\square$

Next, we use Theorem 3.9 to establish that even *simultaneous* RM$^+$ converges in symmetric potential games, as long as the players have the same initialization.

**Corollary 3.10.** *In any symmetric potential game, simultaneous RM$^+$ converges to an $\epsilon$-Nash equilibrium after $O_\epsilon(1/\epsilon^4)$ rounds. In particular, if convergence to the set of CCE happens at a rate of $T^{-(1-\alpha)}$, for some $\alpha \in [0, 1/2]$, the rate of convergence to Nash equilibria is no worse than $T^{-\frac{1-\alpha}{2}}$.*

*Proof.* We will argue that simultaneous RM$^+$, under the same initialization, in a symmetric game is tantamount to running RM$^+$ with respect to the function $\Delta(\mathcal{A}) \ni \boldsymbol{x} \mapsto \Phi(\boldsymbol{x}, \ldots, \boldsymbol{x})$, where $\mathcal{A}_1 = \cdots = \mathcal{A}_n = \mathcal{A}$. We first claim that for any $a \in \mathcal{A}$,

$$\frac{\partial \Phi(\boldsymbol{x}, \ldots, \boldsymbol{x})}{\partial \boldsymbol{x}[a]} = \sum_{i=1}^{n} \frac{\partial \Phi(\boldsymbol{x}_1, \ldots, \boldsymbol{x}_n)}{\partial \boldsymbol{x}_i[a]} \bigg|_{\boldsymbol{x}_1 = \cdots = \boldsymbol{x}_n = \boldsymbol{x}}. \tag{20}$$

By definition, $\Phi(\boldsymbol{x}_1, \ldots, \boldsymbol{x}_n)$ is multilinear. Let us consider a monomial of $\Phi$ of the form

$$m_{\boldsymbol{a}}(\boldsymbol{x}_1, \ldots, \boldsymbol{x}_n) = \prod_{i=1}^{n} \boldsymbol{x}_i[a_i]$$

for some joint action $\boldsymbol{a} = (a_1, \ldots, a_n) \in \mathcal{A}_1 \times \cdots \times \mathcal{A}_n$. Then

$$
\begin{aligned}
\frac{\partial m_{\boldsymbol{a}}(\boldsymbol{x}, \ldots, \boldsymbol{x})}{\partial \boldsymbol{x}[a]} &= \frac{\partial}{\partial \boldsymbol{x}[a]} \left( \prod_{a' \in \mathcal{A}'} (\boldsymbol{x}[a'])^{d(a')} \right) \\
&= \begin{cases} d(a)(\boldsymbol{x}[a])^{d(a)-1} \prod_{a' \in \mathcal{A}' \setminus \{a\}} (\boldsymbol{x}[a'])^{d(a')} & \text{if } a \in \mathcal{A}' \text{ and} \\ 0 & \text{otherwise,} \end{cases}
\end{aligned} \tag{21}
$$

where $\mathcal{A}' = \mathcal{A}'(\boldsymbol{a}) = \{a' \in \mathcal{A} : \exists i \in [n] \text{ such that } a' = a_i\}$ and degrees $d(a') = |\{i \in [n] : a_i = a'\}| \geq 1$ for $a' \in \mathcal{A}'$. Further,

$$
\frac{\partial m_{\boldsymbol{a}}(\boldsymbol{x}_1, \ldots, \boldsymbol{x}_n)}{\partial \boldsymbol{x}_i[a]} = \begin{cases} \prod_{i' \neq i} \boldsymbol{x}_{i'}[a_{i'}] & \text{if } a = a_i \\ 0 & \text{otherwise.} \end{cases}
$$

In particular,

$$
\left. \frac{\partial m_{\boldsymbol{a}}(\boldsymbol{x}_1, \ldots, \boldsymbol{x}_n)}{\partial \boldsymbol{x}_i[a]} \right|_{\boldsymbol{x}_1 = \cdots = \boldsymbol{x}_n} = \mathbb{1}\{a = a_i\} \boldsymbol{x}[a]^{d(a)-1} \prod_{a' \in \mathcal{A}' \setminus \{a\}} (\boldsymbol{x}[a'])^{d(a')}.
$$

As a result,

$$
\left. \sum_{i=1}^n \frac{\partial m_{\boldsymbol{a}}(\boldsymbol{x}_1, \ldots, \boldsymbol{x}_n)}{\partial \boldsymbol{x}_i[a]} \right|_{\boldsymbol{x}_1 = \cdots = \boldsymbol{x}_n} = \mathbb{1}\{a \in \mathcal{A}'\} d(a) \boldsymbol{x}[a]^{d(a)-1} \prod_{a' \in \mathcal{A}' \setminus \{a\}} (\boldsymbol{x}[a'])^{d(a')},
$$

which matches the expression in (21). That is, we have shown that for any $\boldsymbol{a} \in \mathcal{A}_1 \times \cdots \times \mathcal{A}_n$,

$$
\frac{\partial m_{\boldsymbol{a}}(\boldsymbol{x}, \ldots, \boldsymbol{x})}{\partial \boldsymbol{x}[a]} = \left. \sum_{i=1}^n \frac{\partial m_{\boldsymbol{a}}(\boldsymbol{x}_1, \ldots, \boldsymbol{x}_n)}{\partial \boldsymbol{x}_i[a]} \right|_{\boldsymbol{x}_1 = \cdots = \boldsymbol{x}_n}.
$$

Since $\Phi(\boldsymbol{x}_1, \ldots, \boldsymbol{x}_n) = \sum_{\boldsymbol{a} \in \mathcal{A}_1 \times \cdots \times \mathcal{A}_n} \Phi(\boldsymbol{a}) m_{\boldsymbol{a}}(\boldsymbol{x}_1, \ldots, \boldsymbol{x}_n)$, (20) follows by linearity. Moreover, the symmetry assumption concerning the potential $\Phi$ tells us that

$$
\left. \frac{\partial \Phi(\boldsymbol{x}_1, \ldots, \boldsymbol{x}_n)}{\partial \boldsymbol{x}_i[a]} \right|_{\boldsymbol{x}_1 = \cdots = \boldsymbol{x}_n} = \left. \frac{\partial \Phi(\boldsymbol{x}_1, \ldots, \boldsymbol{x}_n)}{\partial \boldsymbol{x}_{i'}[a]} \right|_{\boldsymbol{x}_1 = \cdots = \boldsymbol{x}_n}
$$

for any $i, i' \in [n]$. Combining with (20), we have that for any $a \in \mathcal{A}$,

$$
\frac{\partial \Phi(\boldsymbol{x}, \ldots, \boldsymbol{x})}{\partial \boldsymbol{x}[a]} = n \left. \frac{\partial \Phi(\boldsymbol{x}_1, \ldots, \boldsymbol{x}_n)}{\partial \boldsymbol{x}_i[a]} \right|_{\boldsymbol{x}_1 = \cdots = \boldsymbol{x}_n = \boldsymbol{x}} \quad \forall i \in [n].
$$

Given that $\mathrm{RM}^+$ is scale invariant, we conclude that simultaneous $\mathrm{RM}^+$ on the potential game is equivalent to running $\mathrm{RM}^+$ on the function $\Delta(\mathcal{A}) \ni \boldsymbol{x} \mapsto \Phi(\boldsymbol{x}, \ldots, \boldsymbol{x})$, and the claim follows from Theorem 3.9. $\qquad \square$

We now turn to the analysis of alternating $\mathrm{RM}^+$ in constrained optimization problems over multiple probability simplices. We clarify that when the regret vector is initialized at $\mathbb{R}_{\geq 0}^{\mathcal{A}_i} \ni \boldsymbol{r}_i^{(0)} \neq \boldsymbol{0}$, as assumed below, the first strategy must be defined consistently as $\boldsymbol{x}_i^{(1)} \propto \boldsymbol{r}_i^{(0)}$.

**Corollary 3.11.** *If $u$ is an $L$-smooth function in $\Delta(\mathcal{A}_1) \times \cdots \times \Delta(\mathcal{A}_n)$ with range $u_{\text{range}}$, $\epsilon$-lazy alternating $\mathrm{RM}^+$ initialized at $\boldsymbol{r}_i^{(0)} = \max\{2\sqrt{m_i}, 9\sqrt{m_i}L\}\mathbf{1}$ for each player $i \in [n]$ requires at most $1 + \frac{4n^4 m^2 u_{\text{range}}^2}{\epsilon^4}$ rounds to reach an $\epsilon$-KKT point of $u$.*

*Proof.* By Lemma 3.8, it follows that $\|\boldsymbol{r}_i^{(t)}\|_2 \geq \|\boldsymbol{r}_i^{(0)}\|_2 = \max\{2m_i, 9m_iL\}$. Following the proof of Lemma 3.7, we have

$$
u(\boldsymbol{x}_{i' \leq i}^{(t+1)}, \boldsymbol{x}_{i' > i}^{(t)}) - u(\boldsymbol{x}_{i' < i}^{(t+1)}, \boldsymbol{x}_{i' \geq i}^{(t)}) \geq \frac{1}{2\|\boldsymbol{r}_i^{(t)}\|_1} \left( \mathrm{BRGap}_i(\boldsymbol{x}_i^{(t)}, \boldsymbol{u}_i^{(t)}) \right)^2 \mathbb{1}\{\mathrm{BRGap}_i(\boldsymbol{x}_i^{(t)}, \boldsymbol{u}_i^{(t)}) > \epsilon\}.
$$

As a result, the telescopic summation yields

$$\sum_{t=1}^{T}\sum_{i=1}^{n}\frac{1}{2\|\boldsymbol{r}_i^{(t)}\|_1}\left(\mathsf{BRGap}_i(\boldsymbol{x}_i^{(t)},\boldsymbol{u}_i^{(t)})\right)^2\mathbb{1}\{\mathsf{BRGap}_i(\boldsymbol{x}_i^{(t)},\boldsymbol{u}_i^{(t)})>\epsilon\}\leq u_{\mathsf{range}},$$

Since $\|\boldsymbol{r}_i^{(t)}\|_1 \leq m\sqrt{t}$ for all $i \in [n]$ and $t \in [T]$, it will take at most $1 + \frac{4m^2 u_{\mathsf{range}}^2}{\epsilon^4}$ rounds to converge to a point in which all players have at most an $\epsilon$ best-response gap, which in turn implies that the KKT gap is at most $n\epsilon$. Rescaling $\epsilon$ concludes the proof. $\qquad\square$

To analyze simultaneous $\mathsf{RM}^+$, we adapt Lemma 3.7 as follows.

**Lemma C.10.** *Let $u$ be an $L$-smooth function over $\Delta(\mathcal{A}_1)\times\cdots\times\Delta(\mathcal{A}_n)$. For any $i \in [n]$ and $\boldsymbol{r}_i \in \mathbb{R}_{\geq 0}^{\mathcal{A}_i}$ with $\boldsymbol{r}_i \neq \boldsymbol{0}$, we define $\boldsymbol{x}_i := \boldsymbol{r}_i/\|\boldsymbol{r}_i\|_1$. Further, let $\boldsymbol{r}_i' := [\boldsymbol{r}_i + \nabla_{\boldsymbol{x}_i}u(\boldsymbol{x}) - \langle\boldsymbol{x}_i,\nabla_{\boldsymbol{x}_i}u(\boldsymbol{x})\rangle\mathbf{1}]^+ \neq \boldsymbol{0}$ and $\boldsymbol{x}_i' := \boldsymbol{r}_i'/\|\boldsymbol{r}_i'\|_1$. If $\|\boldsymbol{r}_i'\|_2 \geq \max\{2m_i, 9m_iL\}$ for any $i \in [n]$, then*

$$u(\boldsymbol{x}') - u(\boldsymbol{x}) \geq \frac{1}{2\max_{1\leq i\leq n}\|\boldsymbol{r}_i'\|_1}\sum_{i=1}^{n}\left(\max_{\boldsymbol{x}_i^\star\in\Delta(\mathcal{A}_i)}\langle\boldsymbol{x}_i^\star - \boldsymbol{x}_i,\nabla_{\boldsymbol{x}_i}u(\boldsymbol{x})\rangle\right)^2.$$

*Proof.* Using the quadratic bound for $u$, we have

$$u(\boldsymbol{x}') - u(\boldsymbol{x}) \geq \langle\nabla u(\boldsymbol{x}),\boldsymbol{x}'-\boldsymbol{x}\rangle - \frac{L}{2}\|\boldsymbol{x}-\boldsymbol{x}'\|_2^2$$

$$= \sum_{i=1}^{n}\left(\langle\nabla_{\boldsymbol{x}_i}u(\boldsymbol{x}),\boldsymbol{x}_i'-\boldsymbol{x}_i\rangle - \frac{L}{2}\|\boldsymbol{x}_i-\boldsymbol{x}_i'\|_2^2\right)$$

$$\geq \sum_{i=1}^{n}\left(\frac{1}{\|\boldsymbol{r}_i'\|_1}\|\boldsymbol{r}_i-\boldsymbol{r}_i'\|_2^2 - \frac{L}{2}\|\boldsymbol{r}_i-\boldsymbol{r}_i'\|_1^2\left(\frac{1}{\|\boldsymbol{r}_i\|_1}+\frac{1}{\|\boldsymbol{r}_i'\|_1}\right)^2\right) \qquad (22)$$

$$\geq \sum_{i=1}^{n}\left(\frac{1}{\|\boldsymbol{r}_i'\|_1}\|\boldsymbol{r}_i-\boldsymbol{r}_i'\|_2^2 - \frac{9m_iL}{2\|\boldsymbol{r}_i'\|_1^2}\|\boldsymbol{r}_i-\boldsymbol{r}_i'\|_2^2\right) \qquad (23)$$

$$\geq \frac{1}{2}\sum_{i=1}^{n}\left(\frac{1}{\|\boldsymbol{r}_i'\|_1}\|\boldsymbol{r}_i-\boldsymbol{r}_i'\|_2^2\right), \qquad (24)$$

where (22) uses the one-step improvement property (Lemma C.5) applied for each player $i \in [n]$ and $\boldsymbol{u}_i := \nabla_{\boldsymbol{x}_i}u(\boldsymbol{x})$ together with Lemma C.6; (23) follows from the fact that $\|\boldsymbol{r}_i\|_1 \geq \|\boldsymbol{r}_i'\|_1 - m_i \geq \frac{1}{2}\|\boldsymbol{r}_i'\|_1$ since $\|\boldsymbol{r}_i'\|_1 \geq \|\boldsymbol{r}_i'\|_2 \geq 2m_i$ and $|\langle\boldsymbol{x}_i-\boldsymbol{x}_i',\nabla_{\boldsymbol{x}_i}u(\boldsymbol{x})\rangle| \leq 1$ for all $\boldsymbol{x}_i' \in \Delta(\mathcal{A}_i)$ (per our normalization assumption); and (24) follows from the assumption that $\|\boldsymbol{r}_i'\|_1 \geq 9m_iL$ for each $i \in [n]$. $\qquad\square$

Similarly to Corollary 3.11, we can now show the following concerning simultaneous $\mathsf{RM}^+$.

**Corollary C.11.** *If $u$ is an $L$-smooth function in $\Delta(\mathcal{A}_1)\times\cdots\times\Delta(\mathcal{A}_n)$ with range $u_{\mathsf{range}}$, simultaneous $\mathsf{RM}^+$ initialized at $\boldsymbol{r}_i^{(0)} = \max\{2\sqrt{m_i},9\sqrt{m_i}L\}\mathbf{1}$ for each player $i \in [n]$ requires at most $1 + \frac{4n^2m^2u_{\mathsf{range}}^2}{\epsilon^4}$ rounds to reach an $\epsilon$-KKT point of $u$.*

*Proof.* By Lemma 3.8, it follows that $\|\boldsymbol{r}_i^{(t)}\|_2 \geq \|\boldsymbol{r}_i^{(0)}\|_2 = \max\{2m_i,9m_iL\}$. By Lemma C.10,

$$u(\boldsymbol{x}^{(t+1)}) - u(\boldsymbol{x}^{(t)}) \geq \frac{1}{2n}\frac{1}{\max_{1\leq i\leq n}\|\boldsymbol{r}_i^{(t)}\|_1}\left(\mathsf{KKTGap}(\boldsymbol{x}^{(t)})\right)^2,$$

and the claim follows. $\qquad\square$

Next, we show how to extend the analysis even when the regret vector is initialized at zero, at the cost of incurring a worse dependence on $1/\epsilon$.

**Theorem 3.12.** *If $u$ is an $L$-smooth function in $\Delta(\mathcal{A}_1)\times\cdots\times\Delta(\mathcal{A}_n)$ with range $u_{\mathsf{range}}$, $\epsilon$-lazy alternating (or simultaneous) $\mathsf{RM}^+$ requires at most $O_\epsilon(1/\epsilon^8)$ rounds to reach an $\epsilon$-KKT point of $u$.*

*Proof.* We analyze $\epsilon$-lazy simultaneous $\mathsf{RM}^+$; the alternating version can be treated similarly. If $\|\boldsymbol{r}_i^{(t)}\|_2 \geq \max\{2m_i, 9m_iL\}$ for all players with best-response gap at least $\epsilon$, we have that $u(\boldsymbol{x}^{(t+1)}) - u(\boldsymbol{x}^{(t)})$ is lower bounded by

$$\frac{1}{2\max_i \|\boldsymbol{r}_i^{(t)}\|_1} \sum_{i=1}^{n} \left(\mathsf{BRGap}_i(\boldsymbol{x}_i^{(t)}, \boldsymbol{u}_i^{(t)})\right)^2 \mathbb{1}\{\mathsf{BRGap}_i(\boldsymbol{x}_i^{(t)}, \boldsymbol{u}_i^{(t)}) > \epsilon\}; \tag{25}$$

this follows similarly to Lemma C.10 since players with best-response gap below $\epsilon$ do not update their strategies (under lazy updates). Now, by Lemma 3.8, it follows that the total number of rounds in which there is a player $i$ with at least an $\epsilon$ best-response gap and $\|\boldsymbol{r}_i^{(t)}\|_2 < \max\{2m_i, 9m_iL\} = R_i$ is at most $\sum_{i=1}^{n}(1 + R_i^2/\epsilon^2)$. We will now bound the number of rounds $T'$ it takes to encounter two such rounds. We observe that, between such rounds, we only update players that have at least an $\epsilon$ best-response gap and $\|\boldsymbol{r}_i^{(t)}\|_2 \geq R_i$. By (25), this can continue for at most $1 + 2u_{\mathsf{range}}m\sqrt{T}/\epsilon^2$ rounds, where $T$ is the total number of rounds it takes for all players to have a best-response gap of at most $\epsilon$. That is, $T' \leq 1 + 2u_{\mathsf{range}}m\sqrt{T}/\epsilon^2$. Further, $T \leq T' + T'\sum_{i=1}^{n}(1 + R_i^2/\epsilon^2)$, and the claim follows by solving in terms of $T$ and rescaling $\epsilon$. $\qquad\square$

## C.2 PROOFS FROM SECTION 4

We conclude with the proofs from Section 4. Below (left figure), we provide an illustrative example of the matrix $\mathbf{B}$, defined earlier in (6), for $m = 6$. The plots in Figure 1 are obtained by running simultaneous $\mathsf{RM}$ (left) and alternating $\mathsf{RM}^+$ (right) on this exact game. It is worth pointing out that one could also use a staircase—instead of a spiral—pattern as shown in the right figure.

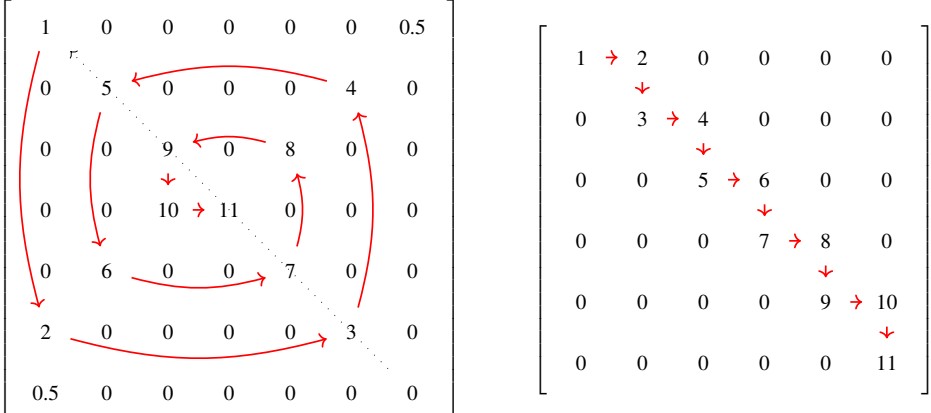

Our main goal is to prove the following invariance.

**Property 4.1.** *After the first round both players play the first action. Thereupon, either the players play with probability $1$ $(a_1(k), a_2(k))$, or, when $k$ is odd, only Player 1 (respectively, Player 2 when $k$ is even) mixes between $a_1(k)$ and $a_1(k+1)$ (respectively, $a_2(k)$ and $a_2(k+1)$). If a row or a column stops being played, it will never be played henceforth. An action profile $(a_1(k+1), a_2(k+1))$ is played with positive probability only if $(a_1(k), a_2(k))$ was played at some previous round.*

It is possible to check the claim for $k = 1, 2, 3, 4$ by executing $\mathsf{RM}$ for a certain number of rounds. In particular, we find that $T_3 \geq 5$ and $T_4 \geq 20$. We proceed by induction in $k$. Suppose that it holds for all payoffs $1, \ldots, \kappa$. We will show that it holds for $\kappa + 1$.

**Lemma C.12.** *For any even $\kappa + 2 \geq k \geq 4$, let $\boldsymbol{r}_1^{(\overline{t_{k-2}})}[a_1]$ be the regret of Player 1 with respect to any action $a_1 \in \mathcal{A}_1(k)$. Then $\boldsymbol{r}_1^{(\overline{t_{k-2}})}[a_1] \leq -\sum_{l=2}^{k-2}(l-1)T_l$. Similarly, for any odd $\kappa + 2 \geq k \geq 5$, if $\boldsymbol{r}_2^{(\overline{t_{k-2}})}[a_2]$ is the regret of Player 2 with respect to any action $a_2 \in \mathcal{A}_2(k)$, $\boldsymbol{r}_2^{(\overline{t_{k-2}})}[a_2] \leq -\sum_{l=2}^{k-2}(l-1)T_l$.*

*Proof.* Let $a_1 \in \mathcal{A}_1(k)$ and $l \in [k-2]$ for an even $k$. Playing $a_1 \in \mathcal{A}_1(k)$ during $[\underline{t_l}, \overline{t_l}]$ gives Player 1 a utility of 0; this follows from the fact that for any column $a_2 \in \{a_2(1), a_2(3), \ldots, a_2(k-3)\} =$

$\{a_2(1), a_2(2), a_2(3), \ldots, a_2(k-3), a_2(k-2)\}$, it holds that $\mathbf{A}[a_1(k), a_2] = 0$, by construction of $\mathbf{A}$. At the same time, Player 1 actually got a utility of at least $l-1$ for each round in $[\underline{t_l}, \overline{t_l}]$. This means that every time Player 1 updates its regret vector within the time period $[\underline{t_l}, \overline{t_l}]$, the regret of $a_1$ decreases by at least $l-1$. The same reasoning applies for Player 2 when $k$ is odd. $\qquad\square$

**Lemma C.13.** *For any even $\kappa \geq k \geq 4$, $T_k \geq -\frac{1}{2} r_2^{(\overline{t_{k-1}})}[a_2(k+1)]$. Similarly, for every odd $\kappa \geq k \geq 5$, $T_k \geq -\frac{1}{2} r_1^{(\overline{t_{k-1}})}[a_1(k+1)]$.*

*Proof.* We fix an even $k \geq 4$. $T_k$ is at least as large as the number of rounds it takes for $a_2(k+1)$ to have nonnegative regret, starting from $r_2^{(\overline{t_{k-1}})}[a_2(k+1)]$. But in every round in $[\underline{t_k}, \overline{t_k}]$ the regret of $a_2(k+1)$ can increase additively by at most 2. This holds because the utility of Player 2 for playing $a_2(k+1)$ is larger than the utility obtained in each round in $[\underline{t_k}, \overline{t_k}]$ by at most 2. So, $T_k \geq \lceil -\frac{1}{2} r_2^{(\overline{t_{k-1}})}[a_2(k+1)] \rceil \geq -\frac{1}{2} r_2^{(\overline{t_{k-1}})}[a_2(k+1)]$. The same reasoning applies when $k$ is odd. $\qquad\square$

The following upper bound on the regret is crude, but will suffice for our purposes.

**Lemma C.14** (Regret upper bound). *For any even $\kappa \geq k \geq 4$, $\|[r_1^{(\overline{t_k})}]^+\|_\infty \leq 2\|[r_1^{(\overline{t_{k-2}})}]^+\|_\infty + 2 \leq \frac{5}{3} 2^{k/2}$ since $\|r_1^{(\overline{t_2})}\|_\infty \leq \frac{4}{3}$. Similarly, for any odd $k \geq 5$, $\|[r_2^{(\overline{t_k})}]^+\|_\infty \leq 2\|[r_2^{(\overline{t_{k-2}})}]^+\|_\infty + 2 \leq \frac{5}{3} 2^{(k-1)/2}$ since $\|r_2^{(\overline{t_3})}\|_\infty \leq \frac{4}{3}$.*

*Proof.* The fact that $\|r_1^{(\overline{t_2})}\|_\infty, \|r_2^{(\overline{t_3})}\|_\infty \leq \frac{4}{3}$ can be shown as part of the basis of the induction. We make the argument for an even $k$. From round $\underline{t_k}$ until Player 1 plays $a_1(k)$ with probability 1, the regret of $a_1(k)$ increases by $k - (kx_1^{(t)}[a_1(k)] + (k-1)x_1^{(t)}[a_1(k-2)]) = x_1^{(t)}[a_1(k-2)]$ and the regret of $a_1(k-2)$ increases by $k - 1 - (kx_1^{(t)}[a_1(k)] + (k-1)x_1^{(t)}[a_1(k-2)]) = -1 + x_1^{(t)}[a_1(k-2)]$; that is, it decreases by $1 - x_1^{(t)}[a_1(k-2)]$. Let $t'$ be the first round for which $r^{(t')}[a_1(k)] \geq r^{(t')}[a_1(k-2)]$. It holds that $r^{(t')}[a_1(k)] \leq \|[r_1^{(\overline{t_{k-2}})}]^+\|_\infty + 1$ since the regret of $a_1(k)$ is increasing by at most 1 in each round and $r^{(t')}[a_1(k-2)] \leq \|[r_1^{(\overline{t_{k-2}})}]^+\|_\infty$. From then onward, the regret of $a_1(k)$ is increasing by at most $1/2$ while the regret of $a_1(k-2)$ is decreasing by at least $1/2$. Thus, it will take at most $\lceil 2|r^{(t')}[a_1(k-2)]|\rceil \leq 2|r^{(t')}[a_1(k-2)]| + 1 \leq 2\|[r_1^{(\overline{t_{k-2}})}]^+\|_\infty + 1$ rounds for the regret of $a_1(k-2)$ to be nonpositive. During that time, the regret of $a_1(k)$ can increase by at most $\|[r_1^{(\overline{t_{k-2}})}]^+\|_\infty + 1$. $\qquad\square$

*Proof of Property 4.1.* If $\kappa$ is odd, it suffices to prove that in every round in which Player 1 mixes between $a_1(\kappa)$ and $a_1(\kappa+1)$, Player 2 plays $a_2(\kappa) = a_2(\kappa+1)$ with probability 1. Similarly, if $\kappa$ is even, it suffices to prove that in every round in which Player 2 mixes between $a_2(\kappa)$ and $a_2(\kappa+1)$, Player 1 plays $a_1(\kappa) = a_1(\kappa+1)$ with probability 1. Let us analyze the case where $\kappa$ is even; the odd case is similar. When Player 2 starts mixing more and more to $a_2(\kappa+1)$, it makes the row $a_1(\kappa+2)$ more attractive for Player 2. By Lemmas C.12 and C.13,

$$r_1^{(\overline{t_\kappa})}[a_1(\kappa+2)] \leq -\frac{\kappa-1}{2} T_\kappa - \frac{\kappa-2}{2} T_{\kappa-1} \leq -\frac{(\kappa-1)!}{2^{\kappa-2}} T_4 - \frac{(\kappa-2)!}{2^{\kappa-3}} T_3. \qquad (26)$$

At the same time, Lemma C.14 implies that Player 2 is mixing between $a_2(\kappa)$ and $a_2(\kappa+1)$ for at most $3\|[r_2^{(\overline{t_{\kappa-1}})}]^+\|_\infty + 2 \leq 5 \cdot 2^{(\kappa-1)/2} + 2$ rounds. To see this, we observe that it takes at most $\lceil \|[r_2^{(\overline{t_{\kappa-1}})}]^+\|_\infty \rceil$ rounds for the action $a_2(\kappa+1)$ to be played with at least the same probability as $a_2(\kappa)$, which in turn holds because the regret of $a_2(\kappa+1)$ increases by $x_2^{(t)}[a_2(\kappa)]$ while the regret of $a_2(\kappa)$ decreases by $1 - x_2^{(t)}[a_2(\kappa)]$. From then on, the regret of $a_2(\kappa)$ decreases by at least $1/2$ in each round, so it takes at most $\lceil 2\|[r_2^{(\overline{t_{\kappa-1}})}]^+\|_\infty \rceil$ rounds for it to be nonpositive. We now claim that, by (26), action $a_1(\kappa+2)$ is never played during those rounds. The reason is that since $T_3 \geq 5$ and $T_4 \geq 20$ (by our inductive basis),

$$r_1^{(\overline{t_\kappa})}[a_1(\kappa+2)] \leq -20\frac{(\kappa-1)!}{2^{\kappa-2}} - 5\frac{(\kappa-2)!}{2^{\kappa-3}}$$

and in each round the regret of $a_1(\kappa + 2)$ can only increase additively by 2. Since

$$\frac{1}{2}\left(20\frac{(\kappa-1)!}{2^{\kappa-2}} + 5\frac{(\kappa-2)!}{2^{\kappa-3}}\right) > 5 \cdot 2^{(\kappa-1)/2} + 2 \quad \forall \kappa \geq 4,$$

the inductive step follows. □

The next lemma shows that, under the invariance of Property 4.1, the only way to reach an approximate Nash equilibrium is to start playing the actions corresponding to $2m - 1$, which is the maximum payoff in the matrix.

**Lemma C.15.** *Consider any strategy profile $(x_1, x_2)$ such that Player 1 only assigns positive probability to actions in $\{a_1(k), a_1(k+1)\}$ and Player 2 only assigns positive probability to actions in $\{a_2(k), a_2(k+1)\}$, where $k+1 < 2m-1$. Then either Player 1 or Player 2 has a deviation benefit of at least $1/k+2 - \gamma$ for any $\gamma > 0$.*

*Proof.* By construction of the game, either $a_1(k) = a_1(k+1)$ or $a_2(k) = a_2(k+1)$. We can assume that $a_1(k) = a_1(k+1)$; the argument when $a_2(k) = a_2(k+1)$ is symmetric. Let $p$ be the probability Player 2 places at $a_2(k+1)$ and $1 - p$ at $a_2(k)$. Suppose that the deviation benefit of each player is at most $\epsilon$. The utility of Player 2 under the current strategy profile is $k(1-p) + (k+1)p = k + p$, while deviating to $a_2(k+1)$ gives $k+1$. So, $p > 1 - \epsilon$. Given that $k + 1 < 2m - 1$, Player 1 can deviate to $a_1(k+2)$ to obtain a utility of $p(k+2) \leq k + p + \epsilon$. Combining with the fact that $p \geq 1 - \epsilon$, this implies $\epsilon \geq 1/k+2$. □

**Uniform initialization** So far, our lower bound assumes that one can initialize RM arbitrarily. A more common initialization prescribes randomizing uniformly at random. In what follows, we point out that the same argument works by considering a different common payoff matrix; namely,

$$\tilde{\mathbf{B}}[a_1, a_2] := \begin{cases} \mathbf{A}[a_1, a_2] & \text{if } 1 \leq a_1 \leq m \text{ and } 1 \leq a_2 \leq m; \\ 1 - \frac{1}{m} & \text{if } a_1 = 1 \text{ and } m+1 \leq a_2 \leq 2m; \\ 1 - \frac{3}{m} & \text{if } m+1 \leq a_1 \leq 2m \text{ and } a_2 = 1; \\ -\frac{1}{m}\sum_{a_2'=1}^{m} \mathbf{A}[a_1, a_2'] & \text{if } 2 \leq a_1 \leq m \text{ and } m+1 \leq a_2 \leq 2m; \\ -\frac{1}{m}\sum_{a_1'=1}^{m} \mathbf{A}[a_1', a_2] & \text{if } m+1 \leq a_1 \leq 2m \text{ and } 2 \leq a_2 \leq m; \\ \frac{1}{m^2}\sum_{a_1'=1}^{m}\sum_{a_2'=1}^{m} \mathbf{A}[a_1', a_2'] - \frac{2}{m} & \text{if } m+1 \leq a_1 \leq 2m \text{ and } m+1 \leq a_2 \leq 2m. \end{cases}$$

By assumption, $x_1^{(1)}$ and $x_2^{(1)}$ are the uniform distributions over $[2m]$. We observe that

$$\sum_{a_1=1}^{2m}\sum_{a_2=1}^{2m} \tilde{\mathbf{B}}[a_1, a_2] = \sum_{a_1=1}^{m}\sum_{a_2=1}^{m} \mathbf{A}[a_1, a_2] + m\left(1 - \frac{1}{m}\right) + m\left(1 - \frac{3}{m}\right)$$

$$- \sum_{a_1=2}^{m}\sum_{a_2=1}^{m} \mathbf{A}[a_1, a_2] - \sum_{a_1=1}^{m}\sum_{a_2=2}^{m} \mathbf{A}[a_1, a_2] + \sum_{a_1=1}^{m}\sum_{a_2=1}^{m} \mathbf{A}[a_1, a_2] - 2m$$

$$= 2\sum_{a_1=1}^{m}\sum_{a_2=1}^{m} \mathbf{A}[a_1, a_2] - \sum_{a_1=2}^{m}\sum_{a_2=1}^{m} \mathbf{A}[a_1, a_2] - 1 - \sum_{a_1=1}^{m}\sum_{a_2=2}^{m} \mathbf{A}[a_1, a_2] - 3 = 0$$

since $1 = \sum_{a_2=1}^{m} \mathbf{A}[a_1 = 1, a_2]$ and $3 = \sum_{a_1=1}^{m} \mathbf{A}[a_1, a_2 = 1]$. In turn, this implies that $x_1^{(1)} \tilde{\mathbf{B}} x_2^{(1)} = \frac{1}{4m^2}\sum_{a_1=1}^{2m}\sum_{a_2=1}^{2m} \tilde{\mathbf{B}}[a_1, a_2] = 0$. Let $u_1^{(1)} = \tilde{\mathbf{B}} x_2^{(1)}$ and $u_2^{(1)} = \tilde{\mathbf{B}}^\top x_1^{(1)}$. We have

$$u_1^{(1)}[a_1] = \begin{cases} \frac{1}{2m}\sum_{a_2=1}^{m} \mathbf{A}[a_1 = 1, a_2] + \frac{1}{2m}m\left(1 - \frac{1}{m}\right) = \frac{1}{2} & \text{if } a_1 = 1; \\ \frac{1}{2m}\sum_{a_2=1}^{m} \mathbf{A}[a_1, a_2] - \frac{1}{2m}m\frac{1}{m}\sum_{a_2=1}^{m} \mathbf{A}[a_1, a_2] = 0 & \text{if } 2 \leq a_1 \leq m; \\ \frac{1}{2m}(1 - 2) = -\frac{1}{2m} & \text{if } m+1 \leq a_1 \leq 2m. \end{cases}$$

Since $\langle x_1^{(1)}, u_1^{(1)} \rangle = 0$,

$$r_1^{(1)}[a_1] = \begin{cases} \frac{1}{2} & \text{if } a_1 = 1; \\ 0 & \text{if } 2 \leq a_1 \leq m; \\ -\frac{1}{2m} & \text{if } m+1 \leq a_1 \leq 2m. \end{cases}$$

Similarly, we have

$$
\boldsymbol{r}_2^{(1)}[a_2] = \begin{cases} \frac{1}{2} & \text{if } a_2 = 1; \\ 0 & \text{if } 2 \le a_2 \le m; \\ -\frac{1}{2m} & \text{if } m + 1 \le a_2 \le 2m. \end{cases}
$$

As a result, the second round sees both players playing the first action with probability 1. In view of the fact that $\mathbf{A}[a_1 = 1, a_2] < 1$ when $1 \le a_2 \le m$ and $\mathbf{A}[a_1, a_2 = 1] < 1$ when $1 \le a_1 \le m$, we immediately revert to the previous analysis concerning simultaneous RM executed on the game described in (6). Thus, we arrive at the following lower bound (by a change of variables $m' \leftarrow 2m$).

**Corollary C.16.** *Simultaneous* RM *requires* $m^{\Omega(m)}$ *rounds to converge to a* $\frac{1}{m+1}$*-Nash equilibrium in two-player* $m \times m$ *identical-interest games. This holds even when both players initialize at the uniform random strategy.*

