# OpenReview forum: "Convergence of Regret Matching in Potential Games and Constrained Optimization"
_ICLR.cc/2026/Conference — ICLR 2026 Poster_

### Official Review · Reviewer_GY2W · 2025-10-25

**Soundness:** 4
**Presentation:** 4
**Contribution:** 3
**Rating:** 8
**Confidence:** 4

**Summary:**

This paper studies regret matching variants, particularly alternating RM and RM+ and explores the convergence of these algorithms to first order optima in constrained optimization problems and potential games. For computing approximate KKT points, the authors show that alternating RM+ converges after $O(1/\epsilon^4)$ iterations,  though a $O(1/\epsilon^2)$ bound can be obtained when the regrets remain bounded. This leverages a novel connection between regret minimization and KKT gaps. Using this result, it is also shown that simultaneous RM+ with symmetric initializations converge to approximate NE with the same complexity as given above. Next, RM is investigated and shown to possess exponential lower bounds, alternation or not. This shows a stark separation between RM and RM+, despite their similarity in terms of the update rule.

**Strengths:**

- The results of this paper are strong, and resolve some open problems in the literature. The separation between RM and RM+ in terms of convergence rate is the first result of its kind.
- The connection between KKT gap and total regret is to my knowledge novel in this setting and results in an intriguing algorithmic consequence of regret accumulation. Future work exploiting this connection would be very interesting indeed.
-  The introductory sections are very clear and readable, even if the technical statements are relatively dense. This makes understanding the ideas of the paper much easier, particularly for readers who are not familiar with the material.

**Weaknesses:**

- While this is not the main point of the paper, perhaps more background and comparison between the convergence of RM+ and other online learning methods like FTRL/Mirror Descent can be added to the main text. This would help readers understand what is known about online learning in potential games, and why the result for alternating RM+ is important. For example, while MWU can exhibit chaotic behavior in potential games, while it seems that alternating RM+ avoids this.
- The reliance on alternation seems to dissuade the practical use-case of learning in a decentralized, simultaneous environment (e.g. where players do not get to observe the strategies used by other players). Having some more results on the simultaneous case would greatly improve the paper. While it is clear why Lemma 3.3 leads to the alternating convergence result, the paper could do with more discussion on where the proof idea fails for the simultaneous case, or if there are further technical problems that arise.

**Questions:**

- Have the authors investigated predictive/smooth predictive RM+ in potential games? Would the $O(1/\epsilon^4)$ convergence remain in this case, or would there be some acceleration?

---

> ### Author Response · Authors · 2025-11-17
>
> Re. *“While this is not the main point of the paper, perhaps more background and comparison between the convergence of RM+ and other online learning methods like FTRL/Mirror Descent can be added to the main text. This would help readers understand what is known about online learning in potential games, and why the result for alternating RM+ is important. For example, while MWU can exhibit chaotic behavior in potential games, while it seems that alternating RM+ avoids this.”*
>
> We have expanded the related work (Appendix A) with references concerning the behavior of other learning algorithms (FTRL or MD) in potential games.
>
> Re. *“Have the authors investigated predictive/smooth predictive RM+ in potential games? Would the  convergence remain in this case, or would there be some acceleration?”*
>
> This is an interesting question. Unlike zero-sum games, we do not believe that predictive variants would lead to faster convergence in this setting. Indeed, unlike zero-sum games, using optimistic gradient descent instead of vanilla gradient descent does not lead to improved rates in the constrained optimization setting, so it’s natural to expect that the same trend would persist for the regret matching family. This is also supported by recent experiments in the paper of Tewolde et al. (2025), where it was found that the predictive variants often perform worse.
>
> We should also point out that analyzing predictive RM(+) seems to require a non-trivial adaptation of our analysis . Lemma 3.8, which is crucial for analyzing RM+ in the general setting, is actually false for predictive RM+ (the paper of Farina et al. (2023) contains an explicit counterexample).

---

> > ### Comment · Reviewer_GY2W · 2025-11-25
> > **Response to rebuttal**
> >
> > Thank you for the explanation and pointing out the recent work on predictive variants in constrained optimization. I remain positive on the paper.

---

### Official Review · Reviewer_bT7s · 2025-10-28

**Soundness:** 3
**Presentation:** 2
**Contribution:** 3
**Rating:** 6
**Confidence:** 4

**Summary:**

This paper investigates the convergence properties of Regret Matching (RM) and its popular variant, Regret Matching+ (RM+), in the context of potential games and, more generally, constrained optimization over a product of simplices. The authors provide two main, contrasting results. First, they establish for the first time that an alternating version of RM+ converges to an approximate KKT point in polynomial time, with a rate of O(1/ε⁴), confirming it is a valid first-order optimization algorithm. Second, they show that standard RM, with or without alternation, can require an exponential number of iterations to converge to even a coarse approximate Nash equilibrium in a simple two-player potential game. This provides the first exponential separation between the performance of RM and RM+, offering strong theoretical justification for the empirical preference for RM+ in practice.

**Strengths:**

1.  The paper tackles a long-standing open problem about the convergence of regret matching in potential games. Establishing the first polynomial-time convergence guarantee for alternating RM+ in this general setting is a major theoretical breakthrough.
2.  The exponential lower bound for standard RM is a very strong result. It clearly and definitively separates RM from RM+, explaining the performance gap often observed empirically. This separation in potential games is novel and fundamentally different from prior work focused on zero-sum games.
3.  The paper is very well-written and structured. The introduction does an excellent job of motivating the problem, and the "Our Results" section clearly lays out the key contributions and their implications. The main theorems are stated precisely, and the narrative guides the reader effectively through the technical arguments.
4.  The connection established between the KKT gap and the accumulated regret is intriguing and provides a novel analytical tool. The proof technique for the RM+ convergence, which handles the lack of a standard one-step improvement property, is clever.

**Weaknesses:**

1.  The proven convergence rate for alternating RM+ is O(1/ε⁴), which is quite slow compared to the O(1/ε²) rate often achievable with standard gradient-based methods for non-convex optimization. While being the first polynomial bound is a great achievement, it would be beneficial to discuss whether this rate is believed to be tight or if it is an artifact of the current analysis.
2. The positive convergence result for simultaneous RM+ is restricted to symmetric potential games under a symmetric initialization. This is a fairly strong assumption and leaves the more general and arguably more practical case of simultaneous updates in arbitrary potential games as an open question.
3.  The paper is entirely theoretical. While the introduction cites recent work showing the strong empirical performance of RM+, the paper would be strengthened by including even a small experiment.

**Questions:**

1.  How do you reconcile the strong empirical performance of RM+ (sometimes outperforming gradient descent, as mentioned in the introduction) with the relatively slow O(1/ε⁴) theoretical convergence rate you establish? Does this suggest that the worst-case instances for RM+ are rare in practice?
2.  What are the primary technical hurdles to extending your convergence proof for simultaneous RM+ from symmetric games to general potential games? Do you suspect that simultaneous RM+ might fail to converge in the general case, or is the analysis just significantly more challenging?
3.  Could you provide some higher-level intuition for why the seemingly minor modification of truncating negative regrets (the key difference between RM+ and RM) prevents the algorithm from getting stalled for an exponential amount of time? The paper proves this formally, but an intuitive explanation would be very helpful.
4.  In Theorem 3.4, the convergence rate for alternating RM+ is parameterized by the growth of the regret. Can you comment on what typical or worst-case regret growth one might expect for RM+ in potential games?

---

> ### Author Response · Authors · 2025-11-17
>
> Re. *“How do you reconcile the strong empirical performance of RM+ (sometimes outperforming gradient descent, as mentioned in the introduction) with the relatively slow $O(1/\epsilon^4)$ theoretical convergence rate you establish? Does this suggest that the worst-case instances for RM+ are rare in practice?”*
>
> This gap between theory and practice mirrors what has been observed in two-player zero-sum games, where RM variants perform very well despite their worst-case complexity relative to other algorithms. This is partly a limitation of a worst-case analysis, as the reviewer suggests.
>
> We should also point out that, in practice, RM+ appears to incur constant regret, in which case our theoretical rate indeed matches the rate of gradient descent. We have not been able to identify an example where RM+ converges at a rate of $\Omega(1/\epsilon^4)$, so we cannot rule out that our analysis can be improved. Pushing over the $1/\epsilon^4$ threshold would require additional ideas on top of our analysis, and is an interesting direction for future work.
>
> Re. *“The paper is entirely theoretical. While the introduction cites recent work showing the strong empirical performance of RM+, the paper would be strengthened by including even a small experiment.”*
>
> The reason we did not include experiments is that there is a previous complementary paper that extensively documents the performance of regret matching variants relative to gradient descent-type algorithms. But we are happy to include a particular experiment the reviewer believes would add value to the paper and is not covered in the earlier paper of Tewolde et al. (2025).
>
> Re. *“What are the primary technical hurdles to extending your convergence proof for simultaneous RM+ from symmetric games to general potential games? Do you suspect that simultaneous RM+ might fail to converge in the general case, or is the analysis just significantly more challenging?”*
>
> As we pointed out in our general response above, our revision shows that simultaneous RM+ enjoys similar convergence bounds to alternating RM+. We hope this addresses this weakness raised by the reviewer.
>
> Re. *“In Theorem 3.4, the convergence rate for alternating RM+ is parameterized by the growth of the regret. Can you comment on what typical or worst-case regret growth one might expect for RM+ in potential games?”*
>
> This is a great question. We have not been able to identify any example in which the regret of RM+ in a potential game grows beyond a constant. Indeed, in practice, it seems that RM+ has constant regret, which further justifies having a result that parameterizes the convergence as a function of the regret. That being said, it’s possible that there is some carefully constructed game in which RM+ does have $\Omega(\sqrt{T})$ regret; this is left as an open question.
>
> Re. *“Could you provide some higher-level intuition for why the seemingly minor modification of truncating negative regrets (the key difference between RM+ and RM) prevents the algorithm from getting stalled for an exponential amount of time? The paper proves this formally, but an intuitive explanation would be very helpful.”*
>
> The main deficiency of RM is that if an action has very negative regret, it will take many iterations for that action to be played, even if it’s the only good action available for that player; this is not so for RM+, by virtue of our one-step improvement lemma. With that in mind, what happens in our lower bound is this: in each period, a player has a single good action. During that period, all other actions accumulate negative regret. Further, in every new period, the good action is different, so each period takes longer and longer, leading to an exponential lower bound. We already have an intuitive explanation of this phenomenon after Theorem 1.4, but we are happy to expand on that if the reviewer sees fit.
>
> Moreover, we have included a figure in Appendix C.2 to convey better the intuition behind the construction.

---

### Official Review · Reviewer_Zjs4 · 2025-10-29

**Soundness:** 3
**Presentation:** 4
**Contribution:** 3
**Rating:** 8
**Confidence:** 4

**Summary:**

This paper studies the behavior of Regret Matching variants in potential games. The two main results are to establish that:
1. Alternating Regret Matching+ (RM+) has last-iterate convergence to Nash at a T^{-1/4} rate.
2. Vanilla Regret Matching (RM), with or without alternation, has exponentially slow last-iterate convergence on a certain identical interest game.

To prove (1), the authors more generally show that alternating RM+ converges to approximate KKT points of (nonconvex) optimization problems over a product of simplices, and they also establish a similar rate of convergence to Nash in *symmetric* potential games under simultaneous RM+.

**Strengths:**

Overall this is a nice work that offers new characterization of the last-iterate convergence properties of regret matching variants beyond zero-sum games. The positive convergence results for (alternating) RM+ for potential games is in contrast to non-convergence results of these algorithms in certain zero-sum games (Lee+2021 and Cai+2025). Establishing the connection between regret and KKT gap is also interesting. In general the paper is well-written and easy to follow.

[Lee+2021]: "Last-iterate Convergence in Extensive-Form Games", NeurIPS 2021

[Cai+2025]: "Last-Iterate Convergence Properties of Regret-Matching Algorithms in Games", ICLR 2025

**Weaknesses:**

There are few areas that could improve the presentation: for example, some additional explanations and fixing or clarifying several incorrect statements (see these listed below in Questions).

**Questions:**

High-level suggestions:
+ It would be helpful to include some additional related work and discussion on the last-iterate convergence behavior of other families of learning algorithms in potential games (e.g., multiplicative weights / FTRL). See for example [Palaiopanos+2017, Cheung+2020, Anagnostides+2022].

High-level questions:
+ Can you offer explanation as to where the proof of Theorem 3.4 fails when considering *simultaneous* RM+? Does the one-step improvement property of Lemma 3.3. still hold? In other words, is alternation necessary for Theorem 3.4 to hold?

+ Similarly, for Corollary 3.10, are symmetric initializations necessary for the result to hold?

+ Do you suspect faster convergence rates are achievable using other RM+ variants like predictive RM+, with or without alternation?

Low-level questions/clarifications:
+ In Proposition 3.1, the left-hand side of the final inequality seems like it should involve the function value of the average-iterate: e.g., $u(\bar x^T) \ge \max_{x} u(x) - R(T)/T$ where $\bar x^T = \sum_{t=1}^T x^t/T$. Similarly, Line 265 should be adjusted to make it clear that, for concave u, vanishing average regret implies convergence of the average-iterate (ergodic convergence), and not last-iterate convergence as is currently written.

+ L338: Do you mean to say that it is open whether RM+ can have $o(\sqrt{T})$ regret in potential games? (we already know RM/RM+ has $O(\sqrt{Tm})$ regret in general).

Other typos:
+ L265: extra superscript parenthesis in $u(x^t)$.
+ L327: seems like bound should read $\le m \sqrt{t}$ (lower-case $t$, not $T$).

-----

[Palaiopanos+2017]: "Multiplicative Weights Update with Constant Step-Size in Congestion Games: Convergence, Limit Cycles and Chaos", NeurIPS 2017.

[Cheung+2020]: "Chaos, Extremism and Optimism: Volume Analysis of Learning in Games", NeurIPS 2020.

[Anagnostides+2022]: "On Last-Iterate Convergence Beyond Zero-Sum Games", ICML 2022.

---

> ### Author Response · Authors · 2025-11-17
>
> Re. *“It would be helpful to include some additional related work and discussion on the last-iterate convergence behavior of other families of learning algorithms in potential games”*
>
> We have updated the related work (Appendix A) to include additional pointers to last-iterate convergence of other algorithms in potential games.
>
> Re. *“Can you offer explanation as to where the proof of Theorem 3.4 fails when considering simultaneous RM+? Does the one-step improvement property of Lemma 3.3. still hold? In other words, is alternation necessary for Theorem 3.4 to hold?”*
>
> As we pointed out in our general response above, our revision shows that simultaneous RM+ enjoys similar convergence bounds to alternating RM+. The analysis is more challenging for simultaneous RM+ because the one-step improvement only holds conditionally on the regrets being sufficiently large, *even in potential games*. But we observed that using Lemma 3.8 one can still show that if the dynamics fail to stabilize, the players will quickly accumulate regret, in which case the one-step improvement kicks in.
>
> Re. *“Do you suspect faster convergence rates are achievable using other RM+ variants like predictive RM+, with or without alternation?”*
>
> This is an interesting question. Unlike zero-sum games, we do not believe that predictive variants would lead to faster convergence in this setting. Indeed, unlike zero-sum games, using optimistic gradient descent instead of vanilla gradient descent does not lead to improved rates in the constrained optimization setting, so one would expect that the same trend would persist for the regret matching family. This is also supported by recent experiments in the paper of Tewolde et al. (2025), where it was found that the predictive variants often perform worse.
>
>
> Re. *“Do you mean to say that it is open whether RM+ can have $o(\sqrt{T})$ regret in potential games?”*
>
> Yes, to our knowledge it is open whether RM+ can do better than $\sqrt{T}$ in potential games.
>
> Re. *“In Proposition 3.1, the left-hand side of the final inequality seems like it should involve the function value of the average-iterate”*
>
> Proposition 3.1 does imply that *almost all* iterates have converged to a near-optimal point, not just the average. That is, for any $\epsilon, \delta > 0$, a $1 - \delta$ fraction of the points satisfy $u(x) \geq \max_{x’} u(x’) - \epsilon$; this follows immediately from Proposition 3.1 since the average regret converges to 0.
>
> Re. *“L327: seems like bound should read $\leq m \sqrt{t}$ (lower-case $t$, not $T$).”*
>
> This is as we intended: we just used the fact that $t \leq T$ to have a uniform bound on the regrets for all $t$.

---

> > ### Comment · Reviewer_Zjs4 · 2025-11-26
> >
> > Thanks to the authors for their reply. I remain positive about the paper.

---

### Official Review · Reviewer_QJyC · 2025-11-02

**Soundness:** 3
**Presentation:** 3
**Contribution:** 3
**Rating:** 6
**Confidence:** 2

**Summary:**

This paper studies the convergence behavior of regret matching (RM) and regret matching+ (RM+) in potential games and general smooth objectives over products of simplices. It proves that alternating RM+ converges to an $\epsilon$-KKT point within $O(1/\epsilon^4)$ iterations and achieves faster rates when per-simplex regret grows slower than $\sqrt{T}$. The work establishes a formal link between accumulated regret and stationarity, showing how no-regret learning can yield approximate equilibrium in non-zero-sum settings.

**Strengths:**

- The paper provides the first general-purpose guarantees for RM+ beyond zero-sum settings. The core technical contribution is a careful connection between KKT gap and accumulated regret, which enables non-asymptotic convergence rates for alternating RM+ in nonconvex constrained optimization

- The algorithmic scope is broad: the results cover any smooth objective over a product of simplices, with potential games as a key special case. Proving $O(1/\epsilon^4)$  convergence to $\epsilon$-KKT for alternating RM+ is a strong and clean statement

- The lower-bound side demonstrates that plain RM can be exponentially slower than RM+ in potential games explains persistent empirical gaps between the two

**Weaknesses:**

- While the $O(1/\epsilon^4)$   guarantee is valuable, it may still be pessimistic relative to practice. The analysis leaves the leading constants implicit and depends on problem parameters (e.g., action-set size, smoothness), which could be large in realistic instance

- The alternating-updates requirement is algorithmically meaningful but differs from the fully simultaneous updates common in large-scale systems

- The theoretical development assumes full-information gradients of a smooth objective on simplices and a normalization of payoff ranges. These are natural for analysis but leave open how the guarantees translate to bandit/partial information, stochastic gradients, or imperfect-recall sampling regimes

**Questions:**

Can you expose the hidden constants in your $O(1/\epsilon^4)$  rate for alternating RM+ and show their dependence on the number of players/actions, Lipschitz constants, and potential range?

Beyond the symmetric setting, how far do your techniques extend to simultaneous RM+? Are there counterexamples where simultaneous RM+ fails to approach $\epsilon$-KKT?

---

> ### Author Response · Authors · 2025-11-17
>
> Re. *“The analysis leaves the leading constants implicit and depends on problem parameters (e.g., action-set size, smoothness), which could be large in realistic instance … Can you expose the hidden constants in your  rate for alternating RM+ and show their dependence on the number of players/actions, Lipschitz constants, and potential range?”*
>
> We only hide the dependencies on problem parameters in our introduction. The main body provides precise bounds that fully specify the dependence on problem parameters for the convergence of RM+. All our bounds have small leading constants, making them relevant in practice.
>
> Re. *“How far do your techniques extend to simultaneous RM+? Are there counterexamples where simultaneous RM+ fails to approach -KKT?”*
>
> As we pointed out in our general response above, our revision shows that simultaneous RM+ enjoys similar convergence bounds to alternating RM+. We hope this addresses this weakness raised by the reviewer.
>
> We also thank the reviewer for bringing up the stochastic/bandit setting! Extending our results in that setting is interesting, and we have mentioned that in the future work of the revision.

---

### Author Response · Authors · 2025-11-17
**Revision summary**

We are grateful to all reviewers for their service and helpful feedback. We have posted a revision that addresses the main comments. Below we briefly explain the key additions in the revision; the individual points and questions of each reviewer are addressed as separate responses.

First, all reviewers pointed out that extending the convergence bounds to simultaneous RM+ would make the paper stronger. We have indeed observed that, using our current techniques, similar convergence bounds can be shown for simultaneous RM+ as well. In particular, we have added two new results in the revision. First, when the regrets can be initialized at a certain constant, simultaneous RM+ converges at the same rate of $O_\epsilon(1/\epsilon^4)$ (Corollary C.11). This follows by directly extending Lemma 3.7 to the case of multiple simplices (Lemma C.10). Second, even when the regrets are initialized at zero, simultaneous RM+ still converges, albeit at a slower rate of $O_\epsilon(1/\epsilon^8)$ (Theorem 3.12); this is a direct consequence of Lemma 3.8. Both of these results are direct implications of the tools present in our submission, which is why we wanted to include them in our revision. We have also adjusted our introduction to point out this extension. We hope this addresses the questions of the reviewers concerning simultaneous RM+.

We also spotted a missing precondition in an earlier theorem statement concerning the initialization of the regrets, which has been corrected in the revision (Corollary 3.11); this doesn’t affect our results.

In terms of the writing, we expanded on the related work section to discuss in more detail prior work concerning the convergence of no-regret algorithms in potential games (Appendix A), and expanded the future research section to highlight the stochastic/bandit setting as a natural next step. We also corrected a typo spotted by a reviewer.

Finally, concerning the lower bound, we added a figure (Appendix C.2) to help convey the intuition of the construction. For completeness, our revision also shows that the same lower bound applies even if the players initialize from the uniform random strategy, which is common in practice (last paragraph in Appendix C.2).

---

### Meta-Review · Area_Chair_5zjm · 2026-01-05

**Summary:**

All the Reviewers are positive about this paper, thus I recommend acceptance. The Reviewers have nevertheless raised some concerns, which the Authors properly addressed in the revised version.

**Reviewer Concerns:**

All the concerns raised by the Reviewers have been adequately addressed by the Authors.

**Reviewer Scores:**

Reviewer QJyC, Score: 6 - I believe that the rebuttal would not have changed the reviewer’s opinion, especially given the other reviews.

Reviewer Zjs4, Score: 8 - I believe that the rebuttal would not have changed the reviewer’s opinion, especially given the other reviews.

Reviewer bT7s, Score: 6 - I believe that the rebuttal would not have changed the reviewer’s opinion, especially given the other reviews.

Reviewer GY2W, Score: 8 - I believe that the rebuttal would not have changed the reviewer’s opinion, especially given the other reviews.

---

### Decision · Program_Chairs · 2026-01-26

Accept (Poster)